# PARAMETER ESTIMATION OF LONG MEMORY STOCHASTIC PROCESSES WITH DEEP NEURAL NETWORKS

## ABSTRACT

We present a pure deep neural network-based approach for estimating long memory parameters of time series models that incorporate the phenomenon of long range dependence. Long memory parameters such as the Hurst exponent are critical in characterizing the long-range dependence, roughness, and self-similarity of stochastic processes. The accurate and fast estimation of these parameters is of paramount importance in various scientific fields, including finance, physics, and engineering. We harnessed efficient process generators to provide high-quality synthetic training data to train 1D Convolutional Neural Network (CNN) and Long Short-Term Memory (LSTM) models. Our neural models outperform conventional statistical methods, even if the latter have neural network extensions. Precision, speed as well as consistency and robustness of the estimators are supported by experiments with fractional Brownian motion (fBm), the Autoregressive Fractionally Integrated Moving Average (ARFIMA) process, and the fractional Ornstein-Uhlenbeck process (fOU). We believe that our work will inspire further research in the application of deep learning techniques for stochastic process modeling and parameter estimation.

## 1 INTRODUCTION

Long-range dependence or long memory has critical importance in the scientific modeling of natural and industrial phenomena. On the one hand, from the field of natural sciences, one can find several applications in climate change (Yuan et al., 2022; Franzke et al., 2015), hydrology (Hurst, 1956), detection of epilepsy (Acharya et al., 2012), DNA sequencing (R.C. Lopes, 2006), networks (Willinger et al., 2001)) or in cybersecurity detecting anomalies (Li, 2006). On the other hand, research on long memory implies achievements in financial mathematics, see for example (Qian & Rasheed, 2004; Eisler & Kertesz, 2006) or (Baillie, 1994) for the application of long memory in volatility modeling. Clearly, the presence of long memory in time series data is a common tenet, turning a great deal of attention to models that are capable of capturing this phenomenon. In most stochastic models the impact of past events on future events has a fast decay, and this way, the effect of observations from the distant past, in terms of forecasting ability, is negligible. When long-range dependence is present in a system, predictions concerning the future require information from the complete history of the process - in contrast to Markovian environments, when the most recent events already contain all the information that is necessary for an optimal forecast. When one models data with long memory, it is a crucial task to estimate model parameters, and classical inference methods are often not applicable in the case of long memory processes. We focus our attention on three stochastic processes that are frequently utilized in modern applied mathematics: the fractional Brownian motion (fBm), the Autoregressive Fractionally Integrated Moving Average (ARFIMA), and the fractional Ornstein-Uhlenbeck (fOU) process. In the case of fBm and fOU we focus on the estimation of the Hurst parameter. The Hurst exponent controls the roughness, self-similarity, and the long-range dependence of fractional Brownian motion paths, and this way also influences the characteristics of derivative processes such as the fractional Ornstein-Uhlenbeck process. With regard to ARFIMA models, the differencing parameter $d$ is our target governing the decay of autocovariances, and thus, the decay of memory in the system.

We propose well-known neural network architectures as general tools for estimating the parameters that characterize long memory. An important feature of our approach is that efficient process generators provide the opportunity to train the models on very large amounts of data. This approach has several advantages. On the one hand, we can achieve a clearly better performance compared to traditional statistical estimation methods, even when supplemented with neural networks. Inference is efficient in terms of speed, even for samples of long sequences, and this makes our approach valuable for practical applications. The learning process results in models with good performance even with a moderate amount of training data, and this can be improved further by using a larger amount of training data. Measurements displayed here support the general applicability of the neural network approach.

In recent years, a number of works, utilizing neural networks, emerged on the estimation of the Hurst parameter of fBM. A part of these uses MLPs, in which case, since input of MLPs are of a fixed size, one of the followings happen: either inference can be performed only on a fixed-length series (Ledesma-Orozco et al., 2011; Han et al., 2020), or inference is done on a set of process specific statistical measures enabling a fixed size input to the neural networks (Kirichenko et al., 2022; Mukherjee et al., 2023). A more general approach is the signature-based method described in (Kidger et al., 2019), which can also be used to estimate fBM Hurst, where the extracted statistical descriptors are processed by an LSTM. In the case of these methods, the hybrid application of statistical descriptors and neural networks brings less improvement compared to our purely neural network solutions. This is reflected in the comparison to classical estimation methods. Another shortcoming in recently published methods is that they do not address the possible limitations caused by scaled inputs. In the case of the fOU and ARFIMA processes, so far, we could not find neural network based parameter estimators in the literature. Nevertheless, we would like to make the remark, that in case of non-fractional Ornstein-Uhlenbeck process, the architecture of the estimator presented by (Wang et al., 2022) is in close proximity to the one presented in this paper.

A further advantage of our method is that it maintains good performance evenly over the entire $[0, 1]$ range of the Hurst parameter. When inferring, our method competes with previous methods in terms of speed. We found that the the proposed neural network estimator is consistent in the sense that when trained on longer sequences, the method becomes more accurate - even when inference was done on sequences of different length sequences (with respect to lengths used in training). In the case of the fOU process, we compared the neural network-based Hurst estimates with a quadratic variation estimators, and our method presented much higher accuracy.

The success of the utilized networks (Sec. 3.3) utmostly stems from a large volume of high-quality training data, manifested with the software that was built around the framework of the so-called isonormal processes (see Nualart & Nualart (2018) for the mathematical background, and Sec. 3.2 on the implementation). The underlying path-generating methodology includes the circulant embedding of covariance matrices and the utilization of fast Fourier transform.

## 2 BACKGROUND

### 2.1 THE FRACTIONAL BROWNIAN MOTION AND THE FRACTIONAL ORNSTEIN-UHLENBECK PROCESS

Let $H \in (0, 1)$. The fractional Brownian motion $fBm(H) := \left(B_t^H\right)_{t \geq 0}$ is a continuous centered Gaussian process with covariance function $\text{cov}\left(B_t^H, B_s^H\right) = \frac{1}{2}\left(|t|^{2H} + |s|^{2H} - |t - s|^{2H}\right)$. Here, $H$ is called the Hurst exponent of the process. Let $FBM(H, n, S, T)$: $(S < T)$ denote the distribution of the $\frac{T - S}{n}$-equidistant realizations of $fBm(H)$ on the time interval $[S, T]$. It can be shown that $\Delta FBM(H, n, S, T) \sim \lambda(H, T-S)\Delta FBM(H, n, 0, 1)$, where $\Delta FBM(H, n, S, T)$ is the sequence of increments, and $\lambda(H, T-S)$ is a scalar. If we want to estimate $H$ from $FBM(H, n, S, T)$ we might want to consider a shift invariant neural network on the increments, since then it will be sufficient to train it only on $FBM(H, n, 0, 1)$. We might also consider the scaled and drifted fBm process $fBm(H, \sigma, \mu) := \left(\sigma B_t^H + \mu t\right)_{t \geq 0}$, $H \in (0, 1), \sigma > 0, \mu \in \mathbb{R}$ which is the fractional counterpart of the so called Bachelier model (Musiela & Rutkowski, 2006). When the network is

also drift invariant, it is still sufficient to train the network on realizations of $FBM(H, n, 0, 1)$ to yield an estimator for the parameter $H$ of $fBm(H, \sigma, \mu)$.

## 2.2 AUTOREGRESSIVE FRACTIONALLY INTEGRATED MOVING AVERAGE

A real valued stochastic process $X_j$, $j \in \mathbb{Z}$ is said to be covariance stationary if $EX_j^2 < \infty$ and $EX_j$ are constant for all $j \in \mathbb{Z}$, and the autocovariance function $Cov(X_j, X_{j+k}) = Cov(X_0, X_k) = \gamma(k)$ is constant in $j$ for all $j, k \in \mathbb{Z}$. For every event $A$ from the sigma field generated by $X_j$, $j \in \mathbb{Z}$ there exists a Borel set $C = C(A)$ such that $A = [(X_1, X_2, ...) \in C]$. The event $A$ is invariant if we have $A = [(X_k, X_{k+1}, ...) \in C]$ for all $k \geq 1$. We say that the stochastic process $X_j$, $j \in \mathbb{Z}$ is ergodic if every invariant event has probability zero or one.

A covariance stationary sequence of random variables $\zeta_j$, $j \in \mathbb{Z}$ is said to form white-noise if $E\zeta_0 = 0$, $\gamma(0) = E\zeta_0^2 < \infty$, and $\gamma(k) = 0$ for all $k \in \mathbb{Z}$, $k \neq 0$.

For $d > -1$ we define the fractional difference operator $\nabla^d = \sum_{k=0}^{\infty} \binom{d}{k}(-B)^{-k}$ where $B$ is the backward shift operator, that is $BX_j = X_{j-1}$, and $\binom{d}{k} = \frac{d!}{k!(d-k)!}$.

For $d \in (-1/2, 1/2)$ the ARFIMA$(0, d, 0)$ process is defined as the solution of the difference equation

$$\nabla^d X_j = \zeta_j, \tag{1}$$

where $\zeta_j$, $j \in \mathbb{Z}$ is a white-noise sequence. It is known that when $\zeta_j$, $j \in \mathbb{Z}$ is ergodic, and $d \neq 0$, there is a unique stationary solution to (1) – see Theorem 7.2.2 in Giraitis et al. (2012).

## 2.3 THE FRACTIONAL ORNSTEIN-UHLENBECK PROCESS

Let $H \in (0, 1)$, $\alpha, \sigma > 0$, $\eta, \mu \in \mathbb{R}$. The fractional Ornstein-Uhlenbeck process $(Y_t)_{t \geq 0}$ is the solution of the following stochastic differential equation:

$$dY_t = -\alpha(Y_t - \mu)\, dt + \sigma\, dB_t^H$$
$$Y_0 = \eta.$$

Let $fOU(\eta, H, \alpha, \mu, \sigma)$ denote the distribution of this process on the Borel $\sigma$-algebra of continuous functions. Note that $\mu$ and $\sigma$ are simply scaling and shifting parameters. Namely, if $Y \sim fOU\big((\eta - \mu)/\sigma, H, \alpha, 0, 1\big)$, then $\sigma Y + \mu \sim fOU(\eta, H, \alpha, \mu, \sigma)$. This means that if we can guarantee the scale and shift invariance of the network, it will be sufficient to train a $H$-estimator on realizations from $fOU\big(\eta, H, \alpha, 0, 1\big)$ to cover the distribution on $fOU(\eta, H, \alpha, \mu, \sigma)$.

## 2.4 BASELINE ESTIMATORS

To provide baseline comparisons to our neural network based results we considered the following estimators. For a more detailed account on the baseline estimators see Section B in the appendix.

Rescaled range analysis consists of calculating the statistics $R/S$. The method considers the rescaled and mean adjusted range of the progressive sum of a sequence of random variables. This quantity, in case of fractional Brownian motion obeys a specific power law asymptotics, and the Hurst parameter of the process can be obtained through utilizing a logarithmic linear regression. The term and concept of stems from multiple works of Harold E. Hurst for a historical account on the methodology see (Graves et al., 2017).

Since the box counting dimension of fractional Brownian motion is $2 - H$, estimating the fractal dimension can provide a tool for estimating the Hurst exponent $H$. Based on the estimation of the $p$-variation of a process, we considered a generalization of the variogram estimator (Gneiting et al., 2012) for estimating the fractal dimension. We also considered Higuchi's method (Higuchi, 1988) which also provides a tool for estimating the box counting dimension of fBm.

We would like to make the remark, that according to Theorem 7.2.1 in Giraitis et al. (2012), for the aoutocovariance of an ARFIMA process, we have $\gamma(k) \approx c_d k^{2d-1}$. Thus, in terms of the decay of autocovariance and memory properties (see Definition 3.1.2 in Giraitis et al. (2012)), the ARFIMA$(0, d, 0)$ process corresponds to a fractional noise with Hurst parameter $H = d + 1/2$.

Also, an ARFIMA process, in an asymptotic sense, has similar spectral properties to that of fractional Brownian motion incremets. On one hand this means, that an ARFIMA process offers a potential way to test estimators calibrated to fractional Brownian motion. On the other hand, it is reasonable to apply the above baseline Hurst parameter estimators for estimating the parameter $d$ of ARFIMA$(0, d, 0)$.

The logarithmic likelihood procedure, dubbed as Whittle's method, involves the optimization of a quantity that compares the estimated spectrum to the actual spectrum of the underlying parametric process. This method, on an algorithmic level, has to be tailored to the subject of inference, by providing, in a preliminary way, the parametric spectral density of the underlying process. This test is widely used for inference on the Hurst parameter, and it is the state of the art method for parameter estimation tasks that target ARFIMA processes.

To estimate the Hurst parameter of a fractional Ornstein-Uhlebeck process, the paper (Brouste & Iacus, 2011a) provides a statistical method (QGV) that is based on building an estimator that compares generalized quadratic variations corresponding to certain filtered versions of the input samples. The method presents consistent and asymptotically Gaussian estimators, and can be considered a state of the art analytical tool regarding Hurst parameter inference on fOU processes.

## 3 METHODS

### 3.1 TRAINING PARADIGM

In contrast to a situation characterized by a limited amount of data, we have the opportunity to leverage synthetic data generators to train our neural network models on a virtually infinite dataset. The loss computed on the most recent training batches simultaneously serves as a validation loss, as each batch comprises entirely new synthetic data. This setup not only ensures the absence of overfitting in the traditional sense but also highlights the sole potential issue associated with this training paradigm: the quality of the process generator itself. If the generator fails to approximate the target distribution effectively, there is a risk of overfitting the generator's inherent errors. Thus, the availability of high-quality process generators is essential.

Our setup to obtain parameter estimators by utilizing generators for given families of stochastic processes is the following. Let $\Theta$ be the set of the possible parameters and let $P$ be the prior distribution on $\Theta$. For a fixed $a \in \Theta$, the generator $G^a$ denotes an algorithm that generates sample paths of a stochastic process, where the sample paths are distributed according to the process distribution $Q_a$. This is deterministic in the sense that every iteration, returns a sample path. This algorithm however can be treated as a random object, by introducing randomness into the parameter $a$ by setting $a = \vartheta$, where $\vartheta$ is a random variable distributed according to some law $P$. Denote the compound generator by $G^{(\vartheta)}$. Now, suppose we have $G^{(\vartheta)}$ as input and we would like to estimate $\vartheta$. Formally, an optimal $M$ estimator would minimize the MSE $E[M(G^{(\vartheta)}) - \vartheta]^2$. By having independent realizations of series from $G^{(\vartheta)}$, we can consider the training set $T$. Training a proper neural network $\mathcal{M}$ on $T$ with the squared loss function would be a heuristic attempt to obtain the above $M$ estimator. We may assume that $Q$ is only parametrized by the target parameter $a$. This can be done without loss of generality, because if $Q$ is parametrized by other parameters besides $a$, then we can just randomize those parameters and have $Q_a$ be redefined as the resulting mixed distribution.

### 3.2 GENERATING FRACTIONAL PROCESSES

To generate the fractional processes fBM and fOU, we employed the circular matrix embedding method belonging to the Davies-Harte procedure family (Davies & Harte, 1987). In the available Python packages (Christopher Flynn, 2020), the original Davies-Harte method for generation is accessible. However, the generation procedure we use is based on Kroese's method (Kroese & Botev, 2013), which we have re-implemented specifically for generating sequences using the most efficient tools within the Python framework. In our implementation, for multiple sequences, we store the covariance structure so that it does not need to be recomputed each time it is needed. Additionally, the modified version of the traditional Cholesky method is available in the implemented package, which yields a similar level of acceleration for generating large quantities of data, comparable to the

currently available solutions. More details on the generation method and our measurements related to its adequacy are in Section E of the Appendix.

### 3.3 NEURAL NETWORK ARCHITECTURE

There are three kinds of invariances that we might require from the network $\mathcal{M}$: shift, scale, and drift invariance. In order to make an fBm Hurst-estimator which works well in practice, we want to rely on all three of the above invariances. We can obtain shift invariance by transforming the input sequence to the sequence of its increments. Differentiating the input this way also turns drift invariance to shift invariance. By performing a standardization on the sequence of increments we can ensure drift and scale invariance. The standardizing phase can also be considered as an additional layer to the network, applying the transformation $x \mapsto (x - \overline{x})/\hat{\sigma}(x)$ to each sequence of increments $x$ in the batch separately, where $\hat{\sigma}(x)$ is the empirical standard deviation over the sequence $x$, and $\overline{x}$ is the arithmetic mean over $x$.

After the optional layers to ensure invariances, the next layers of $\mathcal{M}$ constitute a sequential regressor. This part of the network first transforms the input sequence into a higher dimensional sequence, after which each dimension of the output is averaged out, resulting a vector. Finally, a scalar output is obtained by an MLP (Haykin, 1994). We considered two options for the sequence transformation. $\mathcal{M}_{\text{conv}}$, where the regression is achieved using a multilayer 1D convolution (Kiranyaz et al., 2021), and $\mathcal{M}_{\text{LSTM}}$, where we use an LSTM (Hochreiter & Schmidhuber, 1997). We found that unless $\mathcal{M}$ is severely underparametrized, the specific hyperparameter configuration does not have a significant effect on its performance. Generally $\mathcal{M}_{\text{LSTM}}$ achieves a little better loss than $\mathcal{M}_{\text{conv}}$, while $\mathcal{M}_{\text{conv}}$ is computationally faster than $\mathcal{M}_{\text{LSTM}}$. Slight differences can arise in the speed of convergence, but these are not relevant due to the unlimited data and fast generators. The following are the hyperparameters that we used in the experiments below.

$\mathcal{M}_{\text{conv}}$ utilizes a 1D convolution with 6 layers. The input has 1 channels, and the convolution layers have output channels sizes of 64, 64, 128, 128, 128, and 128. Every layer has stride 1, kernel size 4 and no padding. The activation function is PReLU after each layer. $M_{\text{LSTM}}$ consists of an unidirectional LSTM with two layers, its input dimension is 1, the dimension of its inner representation is 128. In both models, we use an MLP of 3 layers (output dimensions of 128, 64 and 1), with PReLU activation function between the first two layers. AdamW optimization on the MSE loss function was used for training the models (Loshchilov & Hutter, 2017). The learning rate was set to $10^{-4}$ and the train (and validation) batch size to 32.

### 3.4 TECHNICAL DETAILS

The process generators were implemented in Python (Van Rossum & Drake, 2009), using Numpy (Harris et al., 2020) and SciPy (Virtanen et al., 2020). We imported Higuchi's method from the package AntroPy (Vallat, 2022). The R/S method was imported from the package hurst (Mottl, 2019). We generated the ARFIMA trajectories using the arfima package (Kononovicius, 2021). The framework responsible for the training process was implemented in Pytorch (Paszke et al., 2019). Every neural module we used was readily available in Pytorch. We managed our experiments using the experiment tracker (neptune.ai, 2022). The neural models were trained on Nvidia RTX 2080 Ti graphics cards. Training took approximately one GPU hour to one GPU day per model, depending on the type of process, the applied architecture, and on the length of sequences used for training. A shorter training time can mostly be expected from the acceleration (parallelization) of the sequence generation.

## 4 EXPERIMENTS

### 4.1 METRICS

In addition to the standard MSE loss we use two metrics. For a $H$-estimator $M$, let $b_\varepsilon(x) = m_\varepsilon(x) - x$ be the empirical bias function of radius $\varepsilon$, where $m_\varepsilon(x)$ is the average of estimations the estimator $M$ produces for the input sequences with $H \in [x - \varepsilon, x + \varepsilon]$. Similarly, let the empirical standard deviation function $\sigma_\varepsilon(x)$ be defined as empirical standard deviation of the estimations $M$ produces for inputs inside the sliding window of radius $\varepsilon$.

Let us denote the approximate absolute area under the Hurst–bias curve by $\hat{b}_\varepsilon := \varepsilon \sum_{j=0}^{[1/\varepsilon]} |b_\varepsilon(\varepsilon j)|$, and the approximate area under the Hurst–$\sigma$ curve by $\hat{\sigma}_\varepsilon := \varepsilon \sum_{j=0}^{[1/\varepsilon]} \sigma_\varepsilon(\varepsilon j)$. We use these two metrics in addition to MSE, as they highlight different aspects of the estimators performance.

### 4.2 EVALUATING NEURAL FBM HURST ESTIMATORS ON DIFFERENT LENGTH REALIZATIONS

Due to self-similarity properties of fractional Brownian motion, and the stationarity of increments, a standardizing layer in the network architecture enables us to train only on realizations $\Delta FBM(H, n, 0, n)$. This simplified procedure yields an estimator that, in case of inference, is universally efficient regardless of the equidistant time-scale we choose, and regardless of the terminal time of the targeted process. Due to ergodicity properties of the underlying process, using the term standardizing is adequate - regardless of the fact that we have co-dependent data points as inputs.

In order to evaluate the empirical consistency of the fBm estimators we devised two experiments. We generated the sequences for training and evaluation from $FBM(H, n, 0, n)$, where $H \sim U(0, 1)$ is random. In the first experiment we trained the neural models $M_{\text{conv}}$ and $M_{\text{LSTM}}$ on $n = 100$. $M_{\text{conv}}$ and $M_{\text{LSTM}}$ were both working on the increments, with the standardization layer turned on, to ensure scale, shift and drift invariance. This initial training phase was performed on 200 and 100 virtual epochs each containing 100000 sequences for $M_{\text{conv}}$ and $M_{\text{LSTM}}$ respectively. We fine-tuned the initial models on $n = 200, 400, 800, 1600, 3200, 6400$ and $12800$ for an additional 20 and 10 virtual epochs for $M_{\text{conv}}$ and $M_{\text{LSTM}}$ respectively. At the end we got a model which was trained on all of the sequence lengths in the list, albeit not all at the same time. For example the loss of the model fine-tuned for $n = 3200$ was measured after fine tuning the previous version fine-tuned for $n = 1600$. We can see the results in Tables 1–2, and Figure 1 on the left side, and Figure 2. This represents the best-case scenario for the neural models in terms of sequence length, because loss was measured on sequences with the same length as the models were fine-tuned for.

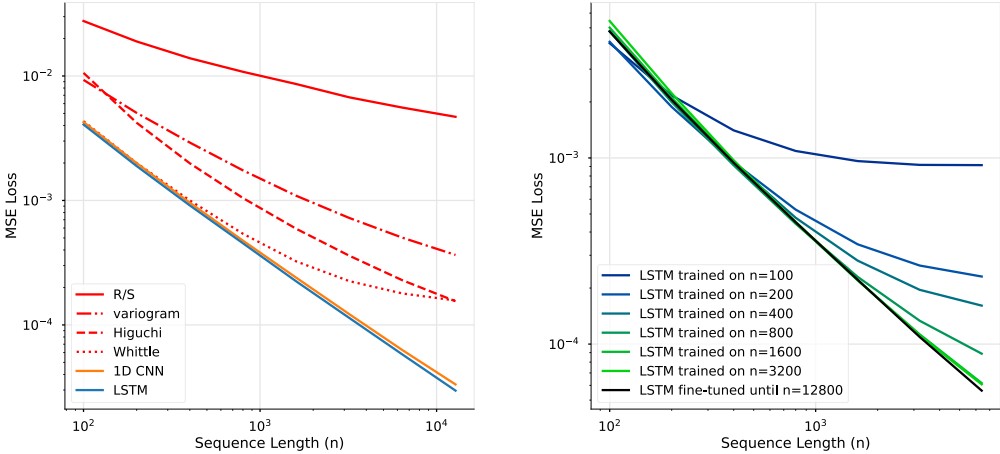

Figure 1: MSE losses of the different estimators by sequence length on a log-log scale. On the left: comparison of the baseline estimators and the fine-tuned neural models. On the right: the empirical consistency of the different dedicated LSTM models.

We also evaluated the empirical consistency of $M_{\text{LSTM}}$ trained exclusively on certain length sequences. We can see the results in Table 2, and Figure 1 on the right side. Trained on shorter sequences the performance of $M_{\text{LSTM}}$ improved when tested on longer sequences, but not as fast as $M_{\text{LSTM}}$ variants trained on longer sequences. $M_{\text{LSTM}}$ variants trained on longer sequences still performed well on shorter sequences, but not ass well as dedicated variants.

In our measurement environment, we found that the neural network estimators have a very good precision-speed trade-off when inferring. The advantage was only partly due to GPU usage. Inference on CPU for 10000 sequences of length 3200 took 1m 32s for $M_{\text{conv}}$ and 5m 43s for $M_{\text{LSTM}}$. GPU inference on 10000 sequences of length 3200 took 10s for $M_{\text{conv}}$ and 9s for $M_{\text{LSTM}}$.

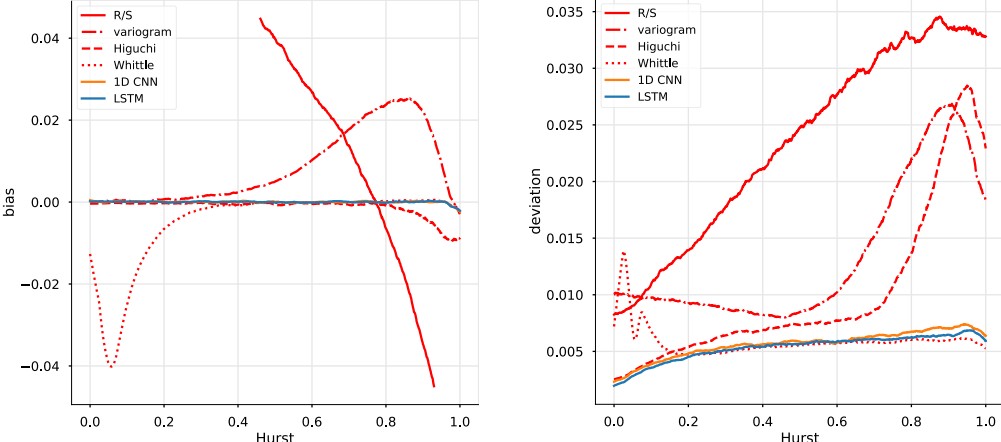

Figure 2: Empirical bias $b_{0.025}$ (on the left) and standard deviation $\sigma_{0.025}$ (on the right) of the fBm estimators by Hurst value. Measured on sequences of length 12800. The $b_{0.025}$ of estimator R/S ranges from 0.125 to -0.075, and was truncated on the plot.

Table 1: MSE losses of different fBm Hurst-estimators by sequence length. To enable direct comparisons with other solutions in the literature, we also included the performance of $M_{\text{LSTM}}$ where only shift invariance is ensured by turning off the standardizing layer ($M_{\text{LSTM}}^*$), here the training and evaluation was performed on $\Delta FBM(H, n, 0, 1)$.

| | | | | $MSE$ loss ($\times 10^{-3}$) | | | |
|---|---|---|---|---|---|---|---|
| seq. len. | R/S | variogram | Higuchi | Whittle | $M_{\text{conv}}$ | $M_{\text{LSTM}}$ | $M_{\text{LSTM}}^*$ |
| 100 | 27.6 | 9.30 | 10.6 | 4.33 | 4.27 | **4.07** | *0.214* |
| 200 | 18.9 | 5.05 | 4.21 | 2.00 | 1.99 | **1.91** | *0.0826* |
| 400 | 13.9 | 2.92 | 1.99 | 1.00 | 0.959 | **0.917** | *0.0366* |
| 800 | 10.8 | 1.75 | 1.05 | 0.540 | 0.476 | **0.453** | *0.0141* |
| 1600 | 8.62 | 1.09 | 0.593 | 0.324 | 0.240 | **0.224** | *0.00715* |
| 3200 | 6.74 | 0.724 | 0.360 | 0.225 | 0.122 | **0.114** | *0.00373* |
| 6400 | 5.57 | 0.502 | 0.229 | 0.179 | 0.0628 | **0.0579** | *0.00646* |
| 12800 | 4.70 | 0.365 | 0.155 | 0.157 | 0.0333 | **0.0297** | *0.00318* |

Table 2: MSE losses of LSTM-based models trained on different sequence lengths.

| train | $MSE$ loss by validation seq. len. ($\times 10^{-3}$) | | | | | | |
|---|---|---|---|---|---|---|---|
| seq. len. | 100 | 200 | 400 | 800 | 1600 | 3200 | 6400 |
| 100 | 4.14 | 2.17 | 1.41 | 1.09 | 0.962 | 0.918 | 0.915 |
| 200 | 4.21 | 1.88 | 0.947 | 0.528 | 0.344 | 0.264 | 0.231 |
| 400 | 4.78 | 2.02 | 0.940 | 0.477 | 0.281 | 0.196 | 0.161 |
| 800 | 4.80 | 2.00 | 0.913 | 0.443 | 0.230 | 0.134 | 0.0888 |
| 1600 | 5.01 | 2.11 | 0.952 | 0.447 | 0.220 | 0.113 | 0.0617 |
| 3200 | 5.44 | 2.23 | 0.972 | 0.454 | 0.221 | 0.111 | 0.0608 |
| 6400 | 5.59 | 2.30 | 1.01 | 0.471 | 0.229 | 0.121 | 0.0692 |

### 4.3 EVALUATING NEURAL ARFIMA PARAMETER ESTIMATORS

We also trained $M_{\mathrm{LSTM}}$ models for estimating the parameter $d$ of the ARFIMA$(0, d, 0)$ process. Here we performed no standardization, and were working with the input sequence, not the increments. We trained $M_{\mathrm{LSTM}}$ on sequences of length 200, 400, 800, 1600, 3200, 6400 and 12800. We evaluated the classical Hurst estimation techniques for the inference of $d$ as described in Section 2.4. We also evaluated Whittle's method calibrated specifically for the ARFIMA $d$-estimation. We can see the results in Table 3, and Figure 3.

We tested the $M_{\mathrm{LSTM}}$ model which was trained on $FBM(H, 12800, 0, 12800)$ on ARFIMA$(0, d, 0)$ trajectories, and an $M_{\mathrm{LSTM}}$ which was trained on ARFIMA$(0, d, 0)$ trajectories of length 12800 on $\Delta FBM(H, 12800, 0, 12800)$ trajectories. As Figure 3 shows, the models perform remarkably — with minor asymmetric bias with respect to the parameter range. This phenomenon attracts a logic that the model either captures the decay rate of autocovariance of fractional noise or some fractal property of sample paths. A number of additional stress tests – where a parameter estimation model trained with a given process was evaluated on a different type of process – can be found in Section D of the Appendix.

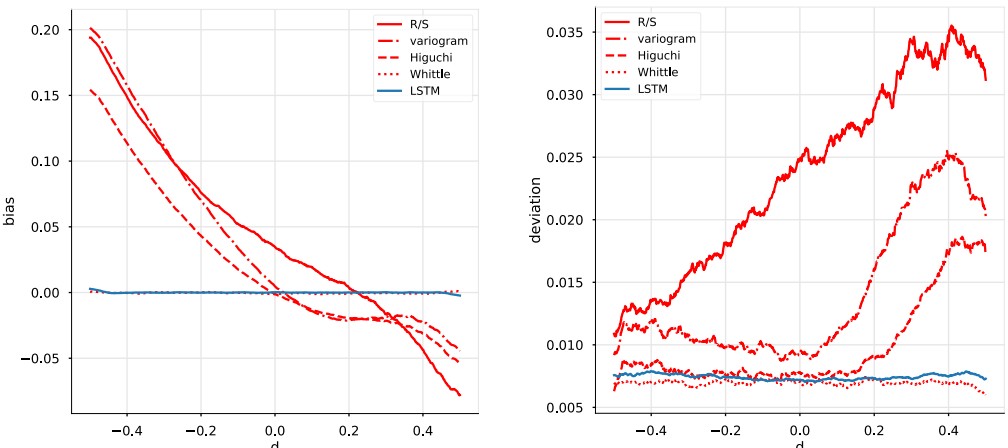

Figure 3: Empirical bias $b_{0.025}$ (on the left) and standard deviation $\sigma_{0.025}$ (on the right) of the ARFIMA$(0, d, 0)$ estimators by $d$. Measured on sequences of length 12800.

Table 3: MSE losses of different ARFIMA$(0, d, 0)$ $d$-estimators by sequence length.

| | $MSE$ loss $(\times 10^{-3})$ | | | | |
|---|---|---|---|---|---|
| seq. len. | R/S | variogram | Higuchi | Whittle | $M_{\mathrm{LSTM}}$ |
| 100 | 33.1 | 17.3 | 14.8 | 9.51 | **5.96** |
| 200 | 24.0 | 12.6 | 8.33 | 4.00 | **3.03** |
| 400 | 18.6 | 9.83 | 5.67 | 1.82 | **1.54** |
| 800 | 14.9 | 8.11 | 4.67 | 0.846 | **0.787** |
| 1600 | 12.6 | 7.70 | 4.24 | 0.401 | **0.390** |
| 3200 | 9.90 | 7.00 | 3.80 | 0.200 | **0.199** |
| 6400 | 8.64 | 6.95 | 3.80 | **0.0960** | 0.104 |
| 12800 | 7.51 | 6.94 | 3.75 | **0.0487** | 0.0552 |

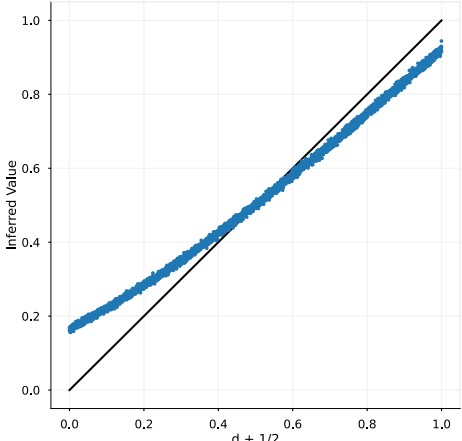 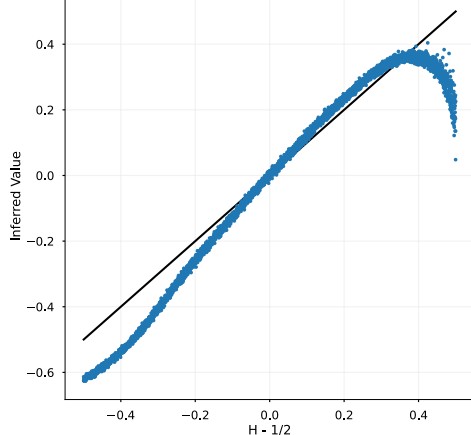

Figure 4: On the left: scatterplot of $M_{\text{LSTM}}$ model finetuned up to 12800 length fBm sequences, inferring on $\text{ARFIMA}(0, d, 0)$ processes of $d \in (-0.5, 0.5)$ and length 12800. On the right: scatterplot of $M_{\text{LSTM}}$ model trained on $\text{ARFIMA}(0, d, 0)$ sequences of length 12800, inferring on $\Delta FBM(H, 12800, 0, 12800)$ sequences.

### 4.4 EVALUATING NEURAL fOU PARAMETER ESTIMATORS

Estimating the parameters of the fractional Ornstein-Uhlenbeck process is significantly more difficult. We evaluated the $M_{\text{LSTM}}$ model on the estimation of the Hurst parameter of fOU. Here $M_{\text{LSTM}}$ was not working with the increments, but standardization was turned on to ensure scale and shift invariance. As we stated in section 2.3 these invariances enable training on $FOU(\eta, H, \alpha, 0, 1)$ without the loss of generality. Additionally we evaluated the quadratic generalized variation (QGV) estimator for the Hurst parameter, as described in (Brouste & Iacus, 2011a). We trained $M_{\text{LSTM}}$ on sequences of length 200, 400, 800, 1600, 3200, 6400 and 12800 with the fine-tuning technique similar to the previous section. We generated the sequences for training and evaluation of the Hurst estimators from $FOU(\eta, H, \alpha, 0, 1)$, where $H \sim U(0, 1), \alpha \sim Exp(100), \eta \sim N(0, 1)$ are random.

Table 4: Performance metrics of different fOU Hurst-estimators by sequence length

|  | $MSE$ loss $(\times 10^{-3})$ | | $\hat{b}_{0.025}$ $(\times 10^{-3})$ | | $\hat{\sigma}_{0.025}$ $(\times 10^{-2})$ | |
| --- | --- | --- | --- | --- | --- | --- |
| seq. len. | QGV | $M_{\text{LSTM}}$ | QGV | $M_{\text{LSTM}}$ | QGV | $M_{\text{LSTM}}$ |
| 100 | 41.0 | **3.38** | 106 | **9.26** | 9.86 | **5.53** |
| 200 | 34.2 | **1.74** | 97.1 | **4.84** | 8.07 | **4.00** |
| 400 | 29.4 | **0.919** | 86.4 | **2.59** | 6.92 | **2.92** |
| 800 | 25.0 | **0.494** | 76.2 | **1.52** | 5.88 | **2.15** |
| 1600 | 20.6 | **0.269** | 65.1 | **0.827** | 5.09 | **1.59** |
| 3200 | 16.3 | **0.149** | 53.8 | **0.575** | 4.37 | **1.18** |
| 6400 | 12.6 | **0.0810** | 43.7 | **1.95** | 3.68 | **0.842** |

## 5 CONCLUSION

In this work, we have demonstrated the utility of sequence-processing neural networks as an effective estimation tool for determining parameters connected to long-memory, e.g. the Hurst exponent. Specifically, we presented the superior performance and consistency of pure neural network-based models with several commonly used techniques in the context of fBm, fOU, and ARFIMA processes. An interpretation of these results is that complex statistical descriptors can emerge as representations by neural networks. As a future endeavor we plan to investigate this phenomenon in greater detail. We believe that the proposed parameter estimators of stochastic processes, based on recurrent neural networks, with their usability and flexibility, form a good basis for the estimation methods that can be used for more intricate processes.

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

## A    APPENDIX - INTRODUCTION

The material in the Appendix is organized in the following way. Firstly, a short description of the conventional numerical-statistical estimation mechanisms used in our experiments is presented in Section B. We add more numerical results from our measurements in Section C.

We want to understand how the neural network-based Hurst parameter estimators trained on e.g. the fBM process perform when applied to other types of processes. From these measurements, we expect to gain insight into what properties of the process the estimator responds to, such as memory, self-similarity, or the fractal dimension of the trajectory. Results of this type of experiment are presented in Section D.

The further part of the measurements supports the proper functioning of the neural parameter esti-mation models. For this, on the one hand, we examine the adequacy of our generator systems used for teaching by performing high-precision statistical tests on the generated realizations (Section E). On the other hand, we also tested the neural estimators on sequences produced with tried-and-tested process generators implemented independently from our project (Section F).

We note that the Python codes of the process generator and the neural network-based estimators will be available.

# B    SUMMARY OF THE BASELINE ESTIMATORS

## B.1    RESCALED RANGE ANALYSIS

The term and concept of rescaled range analysis stems from multiple works of Harold E. Hurst - see e.g. [4], a study in hydrology, and for a historical account on the methodology see (Graves et al., 2017). The statistics $R/S$, defined below, is the rescaled and mean adjusted range of the progressive sum of a sequence of random variables, more precisely, given $Z_1, Z_2, ...$, for a positive integer $n$ consider the statistics

$$R/S(n) = \frac{\max_{1 \le k \le n} \left\{ X_k - \frac{k}{n} X_n \right\} - \min_{1 \le k \le n} \left\{ X_k - \frac{k}{n} X_n \right\}}{\sqrt{\frac{1}{n} \sum_{k=1}^{n} \left( Z_k - \frac{1}{n} X_n \right)^2}}, \quad (2)$$

where $X_k = \sum_{i=1}^{k} Z_i$. The analysis is done via assuming an asymptotics for the statistics (2), namely we postulate that on the long run, it holds that $R/S(n) \approx cn^h$, where $c$ is an unknown constant and $h$ is the parameter we are looking for. Utilizing a one parameter log-log regression on the above formula, that is, using the relation $\log(R/S(n)) = \log(c) + h \log(n)$, one can estimate $h$.

Turning to fractional Brownian motion, it is shown in [6], that its increment process, fractional Gaussian noise has the property $R/S(n) \approx c_0 n^H$, where $H$ is the Hurst parameter of the underlying fractional process, and $c_0$ is some positive constant: yielding a numerical method for the estimation of $H$.

A known limitation of this methodology - when the underlying process is a fractional Brownian motion - is that using the statistics in (2) produces inferred values that are lower when the true value of $H$ is in the long memory range, and substantially higher values when the time series shows heavy anti-persistence – see for example [7]. A possible mitigation of this is to introduce a correction that calibrates the method to fit fractional Brownian motion data and use the corrected estimator as a baseline.

## B.2    VARIATION ESTIMATORS

A generalization of the variogram estimator based on the variogram of order $p$ for a stochastic process with stationary increments is utilized in this paper (Gneiting et al., 2012). The variogram of order $p$ is defined as $\gamma_p(t) = \frac{1}{2} \mathbf{E} |X_i - X_{i+t}|^p$. Notably, when $p = 2$, the variogram is obtained, while for $p = 1$, the madogram is obtained. The case of $p = \frac{1}{2}$ corresponds to the rodogram. In this study, we specifically focus on the case of $p = 1$, where the fractal dimension can be estimated. The fractal dimension is determined using the following formula: $\hat{D}_{V;p} = 2 - \frac{1}{p} \frac{\log \hat{V}_p\left(\frac{2}{n}\right) - \log \hat{V}_p\left(\frac{2}{n}\right)}{\log 2}$. By applying the derived fractal dimension, we can calculate the Hurst exponent ($H$) as $H = 2 - D$(Gneiting & Schlather, 2004).

## B.3    HIGUCHI'S METHOD

Higuchi's method (Higuchi, 1988) relies on the computation of the fractal dimension by a one dimensional box counting. For a sliding box size $b \in \mathbb{N}$ and a starting point $i \in \mathbb{N}, i \le b$, consider

$L_b(i) = \frac{1}{\left[\frac{n-i}{b}\right]} \sum_{k=1}^{\left[\frac{n-i}{b}\right]} \left| X_{i+kb} - X_{i+(k-1)m} \right|$ . Then let $L_b := \frac{1}{b} \sum_{i=1}^{b} L_b(i)$. If $X \sim fBm(H, 0, \sigma)$ then $E(L_b) = cb^H$ holds. Thus, the slope coefficient of the linear regression $\log(L_b) \sim \log(b)$ yields an estimate for $H$.

## B.4    WHITTLE'S METHOD

The likelihood procedure dubbed as Whittle's method, see e.g. (Moran & Whittle, 1951), is based on approximating the Gaussian log-likelihood of a sample of random variables $X = (X_1, ..., X_n)$, where the underlying process is stationary and Gaussian. We give the details in the case when we wish to estimate the Hurst parameter of fractional Brownian motion with Hurst parameter $H \in (0, 1)$, and we apply Whittle's method on its increments. Denoting with $\Gamma_H$ the covariance matrix

corresponding to the vector $X$, the likelihood of the sample with respect to $H$ can be written as $L(X) = (2\pi)^{-n/2} |\Gamma_H|^{-1/2} e^{-\frac{1}{2}X^T\Gamma_H^{-1}X}$, where $|\Gamma_H|, \Gamma_H^{-1}$ denotes the determinant and the inverse of the matrix $\Gamma_H$ respectively, and $X^T$ denotes the transpose of the vector $X$. To speed up the procedure, instead of numerical computations, an approximation can be introduced, see e.g. (Beran, 2017), and the Hurst parameter $H$ can be approximated by minimizing the quantity

$$Q(H) = \int_{-\pi}^{\pi} \frac{I(\lambda)}{f_H(\lambda)} d\lambda, \tag{3}$$

where $I(\lambda)$ is the periodogram, an unbiased estimator of the spectral density $f_H$, defined as $I(\lambda) = \sum_{j=-(n-1)}^{n-1} \hat{\gamma}(j)e^{ij\lambda}$, with the complex imaginary unit $i$, and where the sample autocovariance $\hat{\gamma}(j)$, using the sample average $\bar{X} = \frac{1}{n}\sum_{k=1}^n X_k$, is $\hat{\gamma}(j) = \sum_{k=0}^{n-|j|-1} (X_k - \bar{X})(X_{k+|j|} - \bar{X})$. The quantity in (3) is usually approximated with the sum $\tilde{Q}(H) = \sum_{k=1}^{\lfloor n/2 \rfloor} \frac{I(\lambda_k)}{f_H(\lambda_k)}$, with $\lambda_k = \frac{2\pi k}{n}$, to obtain an asymptotically correct estimate $\hat{H}$ of the Hurst parameter $H$.

## C   Efficiency of models measured in aggregated empirical bias and standard deviations

Table 5: Absolute area under Hurst-bias curve of different fBm Hurst-estimators by sequence length.

| | $\hat{b}_{0.025}$ ($\times 10^{-3}$) | | | | | |
|---|---|---|---|---|---|---|
| seq. len. | R/S | variogram | Higuchi | Whittle | $M_{\text{conv}}$ | $M_{\text{LSTM}}$ |
| 100 | 116 | 34.9 | 58.1 | **10.7** | 12.2 | 11.3 |
| 200 | 98.0 | 26.5 | 27.7 | 6.84 | 5.54 | **5.24** |
| 400 | 87.1 | 20.7 | 14.5 | 5.79 | 2.63 | **2.58** |
| 800 | 79.1 | 17.0 | 7.82 | 5.03 | 1.40 | **1.32** |
| 1600 | 71.4 | 13.9 | 4.63 | 4.90 | 0.765 | **0.656** |
| 3200 | 62.6 | 11.8 | 2.89 | 4.88 | 0.411 | **0.403** |
| 6400 | 58.0 | 9.79 | 1.74 | 5.03 | 0.241 | **0.211** |
| 12800 | 53.7 | 8.34 | 1.23 | 5.04 | 0.140 | **0.131** |

Table 6: Absolute area under Hurst - empirical standard deviation curve of different fBm Hurst-estimators by sequence length.

| | $\hat{\sigma}_{0.025}$ ($\times 10^{-2}$) | | | | | |
|---|---|---|---|---|---|---|
| seq. len. | R/S | variogram | Higuchi | Whittle | $M_{\text{conv}}$ | $M_{\text{LSTM}}$ |
| 100 | 9.62 | 8.67 | 8.25 | 6.25 | 6.16 | **6.01** |
| 200 | 7.35 | 6.33 | 5.65 | 4.30 | 4.27 | **4.15** |
| 400 | 5.69 | 4.78 | 4.04 | 3.01 | 3.00 | **2.91** |
| 800 | 4.58 | 3.61 | 2.97 | 2.11 | 2.12 | **2.05** |
| 1600 | 3.71 | 2.74 | 2.22 | 1.50 | 1.51 | **1.46** |
| 3200 | 3.34 | 2.14 | 1.67 | 1.09 | 1.08 | **1.04** |
| 6400 | 2.76 | 1.67 | 1.28 | 0.791 | 0.775 | **0.743** |
| 12800 | 2.34 | 1.34 | 0.998 | 0.590 | 0.564 | **0.534** |

Table 7: Absolute area under d - bias curve of different ARFIMA(0,d,0) d-estimators by sequence length.

| seq. len. | $\hat{b}_{0.025}$ ($\times 10^{-3}$) | | | | |
|---|---|---|---|---|---|
| | R/S | variogram | Higuchi | Whittle | $M_{\text{LSTM}}$ |
| 100 | 122 | 77.5 | 71.9 | 38.3 | **14.6** |
| 200 | 109 | 68.8 | 57.8 | 19.8 | **7.37** |
| 400 | 98.1 | 63.2 | 50.9 | 10.3 | **3.95** |
| 800 | 90.1 | 58.9 | 48.7 | 5.48 | **2.11** |
| 1600 | 82.7 | 57.8 | 46.9 | 2.70 | **1.17** |
| 3200 | 73.3 | 56.9 | 45.1 | 1.60 | **0.614** |
| 6400 | 68.3 | 56.9 | 44.5 | 0.862 | **0.350** |
| 12800 | 63.6 | 57.4 | 43.9 | 0.495 | **0.216** |

Table 8: Absolute area under d - empirical standard deviation curve of different ARFIMA(0,d,0) d-estimators by sequence length.

| seq. len. | $\hat{\sigma}_{0.025}$ ($\times 10^{-2}$) | | | | |
|---|---|---|---|---|---|
| | R/S | variogram | Higuchi | Whittle | $M_{\text{LSTM}}$ |
| 100 | 9.84 | 8.67 | 8.79 | 8.85 | **7.21** |
| 200 | 7.49 | 6.48 | 5.94 | 5.96 | **5.27** |
| 400 | 5.82 | 4.84 | 4.20 | 4.12 | **3.81** |
| 800 | 4.68 | 3.64 | 3.04 | 2.85 | **2.74** |
| 1600 | 3.83 | 2.77 | 2.22 | **1.98** | **1.98** |
| 3200 | 3.44 | 2.18 | 1.66 | **1.40** | 1.42 |
| 6400 | 2.84 | 1.72 | 1.27 | **0.976** | 1.03 |
| 12800 | 2.37 | 1.40 | 0.980 | **0.696** | 0.744 |

## D  STRESS-TESTING THE NEURAL ESTIMATORS WITH VARIOUS PROCESSES

Assume that we have a black-box estimator denoted by $\Xi$ that receives as input a large number of sample paths $D_H$ of a fractional Brownian motion with unknown parameter $H \in (0, 1)$. The task of the estimator is to reproduce the quantity $H$. If it succeeds, then we can symbolically express this fact with the equality

$$\Xi(D_H) = H.$$

If the above equality holds, one can investigate the unknown methodology that the machine $\Xi$ utilizes to produce its output. The estimator may capture at least three different characteristic quantities: the box dimension of the paths, the properties associated with memory, or the exponent of self-similarity.

### D.1  SUM OF TWO FBMS: BOX DIMENSION VERSUS MEMORY

To differentiate between fractal properties of paths (box dimension) and the decay of auto-covariance (memory), let $H_1 < H_2$ be two real numbers in the set $(0, 1/2) \cup (1/2, 1)$, and let $B_t^{H_1}$ and $B_t^{H_2}$ be two independent fractional Brownian motions with parameters $H_1$ and $H_2$ respectively. Consider the process defined by $X_t = B_t^{H_1} + B_t^{H_2}$. On the one hand, if $\Xi$ captures the asymptotic decay of the autocovariance of the given input, then we have $\Xi[X] = H_2$. On the other hand, contrary to the above case, if $\Xi$ captures the box dimension of the given input, then $\Xi[X] = H_1$. This way, the above heuristic reasoning gives a possible method to test the otherwise unknown behavior of the estimator $\Xi$ regarding its hidden estimation procedure.

Driven by these motivations, after training $M_{\text{LSTM}}$ for the parameter estimation of the fBm, we tested it on the above fBm sums. We considered cases where $H_1$ was fixed and $H_2 \sim U(0, 1)$ was random. The resulting scatter plot can be seen in Figure 5. Apparently, $M_{\text{LSTM}}$ tends to infer values $\hat{H} \in (H_1, H_2)$. Therefore, $M_{\text{LSTM}}$ does not seem to learn the box dimension nor the memory but

an "in-between" quantity. Changing the input sequence length does not change this phenomenon at its core; it only reduces the width of the scatter, i.e., the variance. This confirms our heuristic ideas about the inferral logic of a neural network trained on realizations of the pair $(G, \vartheta)$ (see Subsec. 3.1.). Namely, for an input $x$ the neural network tries to find the expected value $E(\vartheta | G = x)$. It is natural that this obscure conditional expected value does not coincide with any known measurement of memory or dimension.

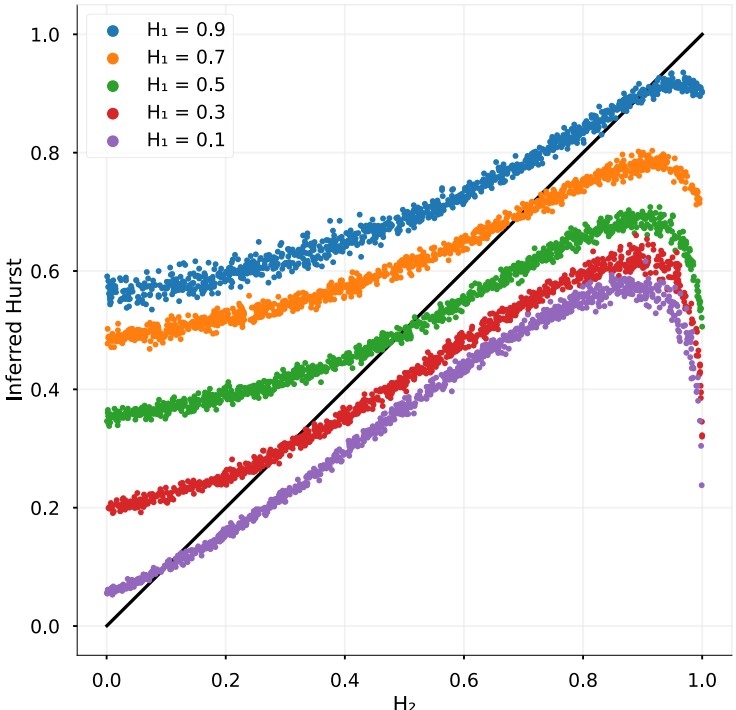

Figure 5: Scatterplot of $M_{\text{LSTM}}$ on fBm sum inputs with fixed $H_1$ and uniform random $H_2$ values. Series length $n = 6400$. $x$: $H_2$, $y$: the sum's $H$ value estimated by $M_{\text{LSTM}}$

### D.2   LÉVY PROCESSES: BOX DIMENSION VERSUS SELF-SIMILARITY

On the one hand, it is a well-known fact that symmetric $\alpha$-stable Levy processes, for $\alpha \in (0, 2]$, are self-similar with self-similarity exponent $1/\alpha$, and according to Seshadri & West (1982) the box dimension of such processes, for $\alpha > 1/2$, can be given by the formula $2 - 1/\alpha$. On the other hand, they do not have memory in the sense that increments are independent. This way, one can assess if the estimator $\Xi$, in case of an $\alpha$-stable Lévy process as input, denoted by $D_\alpha$, produces inferred values according to the law

$$\Xi[D_\alpha] = \frac{1}{\alpha}$$

corresponding to the self-similarity of the underlying, or contrary to this, it rather follows a logic that supports evidence that it infers according to the law

$$\Xi[D_\alpha] = 2 - \frac{1}{\alpha}$$

corresponding to box dimension.

We tested several models, calibrated to fractional Brownian motion, on $\alpha$-stable Levy processes. The results can be seen in Figure 6. In the case of $\alpha = 2$, the law of an $\alpha$-stable Levy process coincides with that of a standard Brownian motion that is also a fractional Brownian motion with

Hurst parameter $H = 1/2$. This way, the inferred value at $\alpha = 2$, that is $1/2$, is not at all unforeseen. However, when moving away from $\alpha = 2$, the model first displays heavy variance, and when moving close to $\alpha = 0$, we see the concentration of inferred values around some levels — that is, as we anticipate, specific to the unique learning phase of the model used.

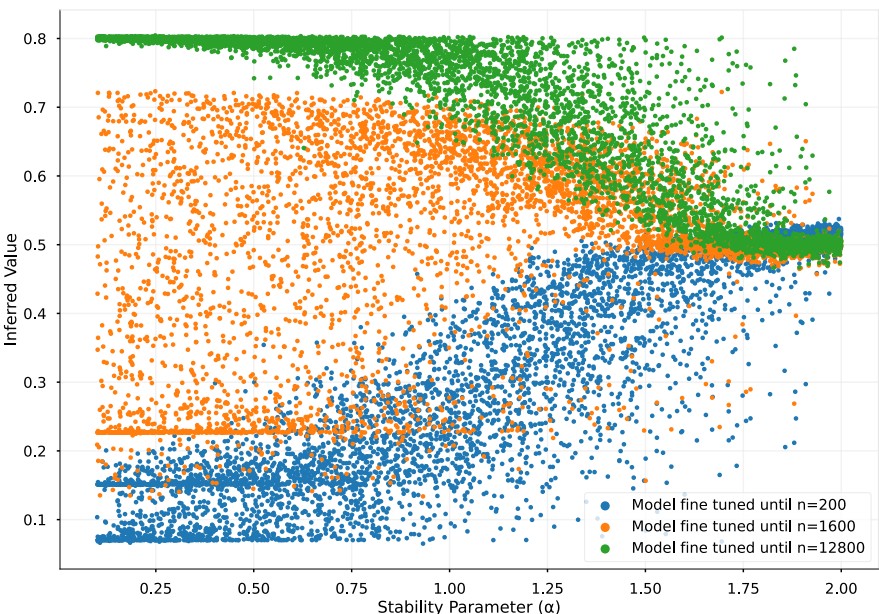

Figure 6: Scatterplot of $M_{\text{LSTM}}$ models finetuned up to different lengths of fBm sequences inferring on Lévy processes of length 6400 and stability parameter $\alpha$.

## D.3 AUTOREGRESSIVE PROCESSES

We tested $M_{\text{LSTM}}$ on autoregressive dynamics of order 1; results are shown in Figure 7. There are two parameter values that can be explained with high confidence. On the one hand, when the speed of mean reversion vanishes, inferred values do so too: this corresponds to the fact that, in some sense, as we approach zero with the Hurst parameter $H$, that is when $H \rightarrow 0$, increments of fractional Brownian motion display a behavior that is comparable to that of white-noise. On the other hand, when the speed of mean reversion is close to $1$, then the autoregressive process coincides with a random walk driven by the underlying noise and, as such, corresponds to standard Brownian motion, that is, a fractional Brownian motion with Hurst parameter $H = 1/2$, which explains the inferred value. The regularity of the inference curve can probably be explained by the continuous nature of neural networks.

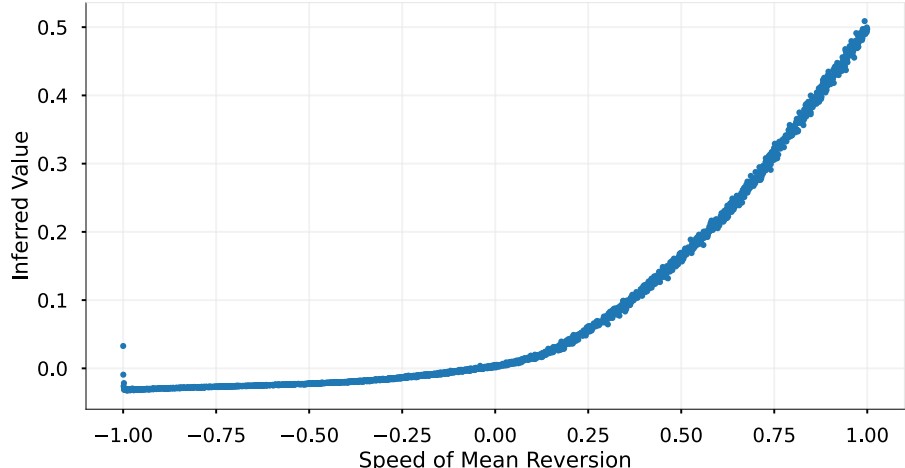

Figure 7: Scatterplot of $M_{\text{LSTM}}$ model finetuned up to 12800 length fBm sequences, inferring on autoregressive processes of order 1 and length 12800.

### D.4 ORNSTEIN–UHLENBECK PROCESSES

We tested $M_{\text{LSTM}}$ on the standard Ornstein-Uhlenbeck process; results are shown in Figure 8. The inferred value at $\alpha = 0$ is $1/2$ as expected — since the model receives an input that it already encountered in the learning phase. We see a decreasing convex curve when the input parameter deviates from zero. A possible explanation is that when $\alpha \neq 0$, the autocovariance shows exponential decay - contrary to the power decay associated with the data the model perceived when calibrated.

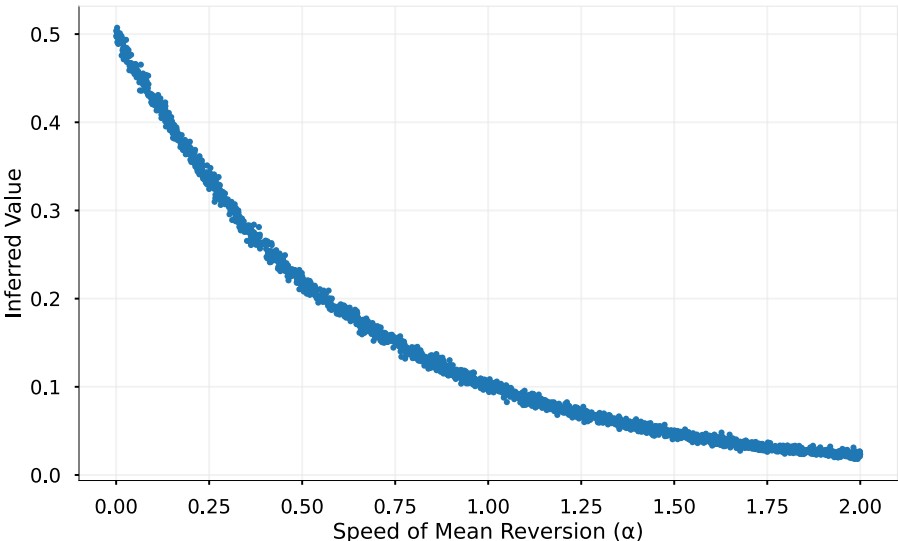

Figure 8: Scatterplot of $M_{\text{LSTM}}$ model finetuned up to 12800 length fBm sequences, inferring on Ornstein–Uhlenbeck processes of length 12800.

# E    STATISTICAL TESTS FOR THE PROCESS GENERATORS

## E.1    DETAILS OF THE GENERATION METHODS

The implemented generation methods can be used for the simulation of sample paths of iso-normal stochastic processes: including fractional integrals (such as fractional Brownian motion, and the fractional Ornstein-Uhlenbeck process). Publication of this work is forthcoming.

Narrowing to the case of fractional Brownian motion, since it is a Gaussian process, a usual methodology can be used for generating trajectories. That is, for a given length of sample path, it is enough to produce a square root decomposition of its covariance matrix over the timestamps. One option is to use the Cholesky method (which can also be used in a more general situation). The other used method utilizes the circulant embedding of the matrix (Davies & Harte, 1987), and fast-Fourier transform (Kroese & Botev, 2013; Dieker & Mandjes, 2003), which to our best knowledge yields paths faster than all known exact methods.

The details are the following. Let us consider the covariance function of the fBm

$$cov(W_t, W_s) = \frac{1}{2} \left( |t|^{2H} + |s|^{2H} - |t-s|^{2H} \right), \quad t, s \geq 0$$

where $H \in (0,1)$. Generating of fBm on an equidistant time grid $0 = t_0 < t_1 < \ldots < t_n = 1$ can be achieved by generating the increment process $(X_1, X_2, \ldots, X_n)$ of the fBm, a fractional Gaussian noise (fGn), where $X_t = W_i - W_{i-1}$. By the cumulative sum of the increment process we have

$$W_{t_i} = c^H \sum_{k=1}^{i} X_k$$

where $i = 1, 2, \ldots, n$ and $c = 1/n$.

The fGn can be characterized as a discrete zero-mean Gaussian process with the following covariance function

$$cov(X_i, X_{i+k}) = \frac{1}{2} \left( |k+1|^{2H} + |k|^{2H} + |k-1|^{2H} \right), \quad k = 0, 1, 2 \ldots$$

.   The fGn is stationary so it can be generated efficiently using the circulant embedding approach. Compute the first row of a symmetric $(n+1) \times (n+1)$ Toeplitz covariance matrix $\Omega_{i+1,j+1} = cov(X_i, X_j)$. Build the $2n \times 2n$ circulant matrix which embeds $\Omega$ in the upper left $(n+1) \times (n+1)$ corner. Let $\mathbf{r} = (r_1, \ldots, r_{n+1}, r_n, r_{n+1}, \ldots, r_2)$ the first row of the circulant matrix and $\lambda$ is the one-dimensional FFT of $\mathbf{r}$ defined as the linear transformation $\lambda = F\mathbf{r}$ with $F_{j,k} = \exp\left( \frac{-2\pi ijk}{\sqrt{2n}} \right), j, k = 0, 1, \ldots, 2n - 1$. The real and the imaginary parts of the first $n+1$ components of $F^* diag(\sqrt{\lambda})\mathbf{Z}$, where $\mathbf{Z}$ is a $2n+1$ complex-valued Gaussian vector, yields two independent fractional Gaussian realizations.

## E.2    SPEED OF THE GENERATOR

Table 9: Running times of generating fBm sequences of length $10^5$, first 1 then 100 sequences.

| Method | Time for 1 seq (s) | Time for 100 seqs (s) |
|---|---|---|
| Cholesky (Christopher Flynn, 2020) | 21.5s | 22.8s |
| Cholesky | 5.8s | 7.14s |
| FFT (Christopher Flynn, 2020) | 48.4ms | 2.99s |
| FFT v1 | 17.7ms | 224ms |
| FFT v2 | 5.80ms | 547ms |

In our first implementation of Kroese's method, denoted by FFT v1, the covariance structure is stored in objects, which slows down the execution for the first sequence since object initialization takes time. To compare, in the case of our implementation FFT v2, there are no initialization tasks

involved resulting in a faster generation of only one sequence. As FFT v1 does not need to re-calculate the covariance structure for each sequence, it can generate a large number of sequences significantly faster. Table 9 shows the comparison in speed of the available Python (Christopher Flynn, 2020) and our new implementations.

### E.3   FBM TESTS

In the case of fractional Brownian motion, validation of the generated data can take place via estimation of the Hurst exponent. Among the various available statistical tools, fractal dimension-based estimators proved to be the most accurate: we employed Higuchi's method and p-Variation estimators. Both procedures estimate the fractal dimension of sample paths; thus, utilizing the relationship between this quantity and the Hurst exponent, these methods simultaneously provide estimators for the Hurst parameter.

We generated 1000 sequences to assess the parameter compliance of our fBm generator, both in the Higuchi and variogram cases. The sequence lengths were 100, 200, 400, 800, 1600, 3200, 6400, and 12800, respectively. We divided the Hurst exponent (0.1) range into intervals of 0.1, ranging from 0.1 to 0.9, and included the values 0.01 and 0.99 as well.

Higuchi's method Higuchi (1988) relies on the computation of the fractal dimension by a one-dimensional box counting. For sample size $n$, a sliding box size $b \in \mathbb{N}$, and a starting point $i \in \mathbb{N}, i \leq b$, consider $L_b(i) = \frac{1}{\left[\frac{n-i}{b}\right]} \sum_{k=1}^{\left[\frac{n-i}{b}\right]} \left|X_{i+kb} - X_{i+(k-1)b}\right|$ . Then let $L_b := \frac{1}{b} \sum_{i=1}^{b} L_b(i)$. If $X \sim FBM(H, 0, \sigma)$ then $E(L_b) = cb^H$ holds. Thus, the slope coefficient of the linear regression $\log(L_b) \sim \log(b)$ yields an estimate for $H$.

Table 10: Estimation of the average Hurst exponent from 1000 sequences

| Hurst | Sequence Length | | | | | | | |
|---|---|---|---|---|---|---|---|---|
| | 100 | 200 | 400 | 800 | 1600 | 3200 | 6400 | 12800 |
| 0.01 | −0.0376 | −0.0108 | 0.0011 | 0.0051 | 0.0078 | 0.0085 | 0.0094 | 0.0096 |
| 0.1 | 0.0520 | 0.0777 | 0.0885 | 0.0956 | 0.0979 | 0.0990 | 0.0994 | 0.0999 |
| 0.2 | 0.1561 | 0.1796 | 0.1884 | 0.1958 | 0.1983 | 0.1987 | 0.1993 | 0.1997 |
| 0.3 | 0.2468 | 0.2766 | 0.2896 | 0.2956 | 0.2965 | 0.2990 | 0.2989 | 0.2994 |
| 0.4 | 0.3495 | 0.3750 | 0.3888 | 0.3932 | 0.3972 | 0.3984 | 0.3992 | 0.3995 |
| 0.5 | 0.4484 | 0.4740 | 0.4895 | 0.4939 | 0.4977 | 0.4986 | 0.4991 | 0.4998 |
| 0.6 | 0.5440 | 0.5742 | 0.5860 | 0.5952 | 0.5967 | 0.5985 | 0.5994 | 0.5997 |
| 0.7 | 0.6411 | 0.6711 | 0.6864 | 0.6918 | 0.6968 | 0.6982 | 0.6989 | 0.6996 |
| 0.8 | 0.7335 | 0.7697 | 0.7833 | 0.7902 | 0.7934 | 0.7974 | 0.7979 | 0.7991 |
| 0.9 | 0.8219 | 0.8563 | 0.8755 | 0.8816 | 0.8896 | 0.8904 | 0.8953 | 0.8970 |
| 0.99 | 0.9191 | 0.9504 | 0.9611 | 0.9699 | 0.9726 | 0.9758 | 0.9776 | 0.9785 |

Table 11: Std. of the estimation of the Hurst exponent from 1000 sequences

| Hurst | Sequence Length | | | | | | | |
|---|---|---|---|---|---|---|---|---|
| | 100 | 200 | 400 | 800 | 1600 | 3200 | 6400 | 12800 |
| 0.01 | 0.0310 | 0.0213 | 0.0151 | 0.0097 | 0.0072 | 0.0049 | 0.0036 | 0.0024 |
| 0.1 | 0.0491 | 0.0336 | 0.0231 | 0.0168 | 0.0123 | 0.0084 | 0.0060 | 0.0040 |
| 0.2 | 0.0658 | 0.0449 | 0.0320 | 0.0222 | 0.0151 | 0.0107 | 0.0076 | 0.0054 |
| 0.3 | 0.0762 | 0.0516 | 0.0362 | 0.0257 | 0.0181 | 0.0126 | 0.0090 | 0.0064 |
| 0.4 | 0.0827 | 0.0586 | 0.0385 | 0.0283 | 0.0192 | 0.0135 | 0.0095 | 0.0066 |
| 0.5 | 0.0878 | 0.0599 | 0.0413 | 0.0291 | 0.0200 | 0.0146 | 0.0101 | 0.0071 |
| 0.6 | 0.0953 | 0.0632 | 0.0430 | 0.0303 | 0.0217 | 0.0149 | 0.0104 | 0.0072 |
| 0.7 | 0.0973 | 0.0711 | 0.0476 | 0.0344 | 0.0250 | 0.0174 | 0.0128 | 0.0089 |
| 0.8 | 0.1067 | 0.0716 | 0.0521 | 0.0397 | 0.0310 | 0.0228 | 0.0177 | 0.0143 |
| 0.9 | 0.1028 | 0.0726 | 0.0569 | 0.0455 | 0.0378 | 0.0323 | 0.0277 | 0.0251 |
| 0.99 | 0.0706 | 0.0507 | 0.0422 | 0.0365 | 0.0324 | 0.0288 | 0.0264 | 0.0244 |

In Gneiting et al. (2012), for a stochastic process with stationary increments, based on the variogram of order $p$, a generalization of the p-variation estimators is utilized. The variogram of order $p$ is defined as $\gamma_p(t) = \frac{1}{2}\mathbf{E}|X_i - X_{i+t}|^p$. Notably, when $p = 2$, the variogram is obtained, while $p = 1$ yields the madogram. The case of $p = \frac{1}{2}$ corresponds to the rodogram. In this study, we specifically focus on the case of $p = 1$, where the fractal dimension can be estimated using the formula $\hat{D}_{V;p} = 2 - \frac{1}{p}\frac{\log \hat{V}_p\left(\frac{2}{n}\right) - \log \hat{V}_p\left(\frac{2}{n}\right)}{\log 2}$. By applying the derived fractal dimension, we can calculate the Hurst exponent ($H$) as $H = 2 - D$ Gneiting & Schlather (2004).

Table 12: Estimation of the average Hurst exponent from 1000 sequences

| Hurst | Sequence Length | | | | | | | |
|---|---|---|---|---|---|---|---|---|
| | 100 | 200 | 400 | 800 | 1600 | 3200 | 6400 | 12800 |
| 0.01 | 0.0214 | 0.0154 | 0.0125 | 0.0124 | 0.0114 | 0.0105 | 0.0091 | 0.0102 |
| 0.1 | 0.1280 | 0.1037 | 0.1044 | 0.1070 | 0.1019 | 0.1009 | 0.1012 | 0.1002 |
| 0.2 | 0.2224 | 0.2191 | 0.2083 | 0.2078 | 0.2040 | 0.2019 | 0.2010 | 0.2004 |
| 0.3 | 0.3330 | 0.3218 | 0.3115 | 0.3083 | 0.3039 | 0.3028 | 0.3017 | 0.3012 |
| 0.4 | 0.4455 | 0.4284 | 0.4177 | 0.4146 | 0.4095 | 0.4060 | 0.4038 | 0.4025 |
| 0.5 | 0.5500 | 0.5341 | 0.5282 | 0.5201 | 0.5153 | 0.5105 | 0.5073 | 0.5050 |
| 0.6 | 0.6611 | 0.6453 | 0.6347 | 0.6292 | 0.6220 | 0.6158 | 0.6134 | 0.6101 |
| 0.7 | 0.7549 | 0.7476 | 0.7403 | 0.7344 | 0.7307 | 0.7254 | 0.7208 | 0.7179 |
| 0.8 | 0.8442 | 0.8434 | 0.8376 | 0.8342 | 0.8321 | 0.8303 | 0.8281 | 0.8247 |
| 0.9 | 0.9159 | 0.9184 | 0.9211 | 0.9194 | 0.9214 | 0.9204 | 0.9220 | 0.9219 |
| 0.99 | 0.9811 | 0.9826 | 0.9829 | 0.9847 | 0.9844 | 0.9853 | 0.9859 | 0.9862 |

Table 13: Std. of the estimation of the Hurst exponent from 1000 sequences

| | Sequence Length | | | | | | | |
|---|---|---|---|---|---|---|---|---|
| Hurst | 100 | 200 | 400 | 800 | 1600 | 3200 | 6400 | 12800 |
| 0.01 | 0.1099 | 0.0823 | 0.0558 | 0.0403 | 0.0283 | 0.0197 | 0.0147 | 0.01 |
| 0.1 | 0.1087 | 0.0772 | 0.0535 | 0.0397 | 0.0268 | 0.0197 | 0.0138 | 0.0096 |
| 0.2 | 0.104 | 0.0733 | 0.0522 | 0.0379 | 0.0255 | 0.0182 | 0.0129 | 0.0092 |
| 0.3 | 0.0972 | 0.069 | 0.0509 | 0.0353 | 0.0242 | 0.0179 | 0.0123 | 0.0089 |
| 0.4 | 0.0891 | 0.0645 | 0.0463 | 0.0337 | 0.0227 | 0.0164 | 0.0114 | 0.0084 |
| 0.5 | 0.0841 | 0.0648 | 0.0448 | 0.0329 | 0.0231 | 0.0169 | 0.0120 | 0.0081 |
| 0.6 | 0.0829 | 0.0607 | 0.0459 | 0.0338 | 0.0246 | 0.0187 | 0.0138 | 0.0104 |
| 0.7 | 0.0763 | 0.0601 | 0.0471 | 0.0368 | 0.0311 | 0.0246 | 0.0190 | 0.0152 |
| 0.8 | 0.0724 | 0.0579 | 0.0475 | 0.0411 | 0.0351 | 0.0298 | 0.0251 | 0.0225 |
| 0.9 | 0.0630 | 0.0505 | 0.0427 | 0.0373 | 0.0337 | 0.0305 | 0.0278 | 0.0261 |
| 0.99 | 0.0360 | 0.0302 | 0.0264 | 0.0230 | 0.0219 | 0.0201 | 0.0187 | 0.0176 |

It can be observed that the estimation results are fairly similar for both methods, although the Higuchi method provided slightly more accurate results, while the first-order variogram performs better at the edges. It is also worth noticing that the length of the generated sequence influences the accuracy of these estimations.

### E.4 FOU TEST

We apply a three-step statistical analysis for testing the data quality generated from the fOU implementation. Assuming in real-life scenarios that a time series follows a fractional Ornstein-Uhlenbeck process, we require knowledge of all the parameters. During the test, the first step involves utilizing a generalized quadratic variation-based estimation to determine the Hurst exponent of the fractional process. Brouste & Iacus (2011b)

$$\hat{H}_N = \frac{1}{2} \log_2 \frac{V_{N,\mathbf{a}^2}}{V_{N,\mathbf{a}}}$$

where $\mathbf{a}$ is a discrete filter and

$$V_{N,\mathbf{a}} = \sum_{i=0} N - K \left( \sum_{k=0}^{K} a_k X_{i+k} \right)^2$$

is the generalized quadratic variations associated to the filter $\mathbf{a}$.

Table 14: Estimation of the average Hurst exponent from 1000 sequences

| | | Sequence Length | | | | | | | |
|---|---|---|---|---|---|---|---|---|---|
| Hurst | $\alpha$ | 400 | 800 | 1600 | 3200 | 6400 | 12800 | 25600 | 51200 |
| 0.1 | 0.1 | 0.0538 | 0.1257 | 0.0252 | 0.1184 | 0.1096 | 0.1086 | 0.1035 | 0.0966 |
| 0.2 | 0.2 | 0.1962 | 0.2378 | 0.2050 | 0.1972 | 0.2059 | 0.2098 | 0.2061 | 0.1980 |
| 0.3 | 0.3 | 0.2891 | 0.2592 | 0.2960 | 0.2974 | 0.2988 | 0.3143 | 0.3125 | 0.2983 |
| 0.4 | 0.4 | 0.3798 | 0.3310 | 0.3678 | 0.4087 | 0.3928 | 0.3860 | 0.3954 | 0.4000 |
| 0.5 | 0.5 | 0.5379 | 0.5540 | 0.5041 | 0.4724 | 0.4729 | 0.5048 | 0.5043 | 0.4943 |
| 0.6 | 0.6 | 0.5986 | 0.5628 | 0.5877 | 0.6015 | 0.6001 | 0.6029 | 0.6085 | 0.5941 |
| 0.7 | 0.7 | 0.5983 | 0.7274 | 0.7177 | 0.7090 | 0.7018 | 0.7060 | 0.6958 | 0.7001 |
| 0.8 | 0.8 | 0.7940 | 0.8031 | 0.7212 | 0.8262 | 0.7959 | 0.7915 | 0.8051 | 0.8007 |
| 0.9 | 0.9 | 0.9458 | 0.8666 | 0.9109 | 0.8967 | 0.8996 | 0.9127 | 0.8925 | 0.9047 |

In the second step, we estimate the parameter $\hat{\sigma}$ using the following formula:

$$\hat{\sigma}_N = \left( 2 \cdot \frac{-V_{N,\mathbf{a}}}{\sum_{k,l} a_k a_l |k-l|^{2\hat{H}_N} \Delta_N^{2\hat{H}_N}} \right)^{\frac{1}{2}} \quad .$$

Table 15: Estimation of the average diffusion ($\sigma$) from 1000 sequences

| Hurst | $\alpha$ | Sequence Length | | | | | | | |
|---|---|---|---|---|---|---|---|---|---|
| | | 400 | 800 | 1600 | 3200 | 6400 | 12800 | 25600 | 51200 |
| 0.1 | 0.1 | 0.8272 | 1.1346 | 0.7195 | 1.1097 | 1.0479 | 1.0529 | 1.0300 | 0.9849 |
| 0.2 | 0.2 | 0.9614 | 1.1860 | 1.0269 | 0.9965 | 1.0174 | 1.0403 | 1.0372 | 0.9910 |
| 0.3 | 0.3 | 0.9512 | 0.8083 | 0.9920 | 0.9996 | 1.0130 | 1.0751 | 1.0700 | 0.9925 |
| 0.4 | 0.4 | 0.8966 | 0.7102 | 0.8638 | 1.0516 | 0.9697 | 0.9341 | 0.9748 | 0.9963 |
| 0.5 | 0.5 | 1.1722 | 1.3135 | 1.0419 | 0.8775 | 0.8900 | 1.0193 | 1.0229 | 0.9680 |
| 0.6 | 0.6 | 0.9960 | 0.8165 | 0.9227 | 1.0228 | 1.0062 | 1.0118 | 1.0534 | 0.9703 |
| 0.7 | 0.7 | 0.6046 | 1.1819 | 1.0958 | 1.0521 | 1.0173 | 1.0301 | 0.9723 | 1.0129 |
| 0.8 | 0.8 | 0.9786 | 0.9846 | 0.6245 | 1.1918 | 0.9740 | 0.9600 | 1.0389 | 1.0090 |
| 0.9 | 0.9 | 1.5738 | 0.7436 | 1.1043 | 0.9756 | 1.0040 | 1.1355 | 0.9389 | 1.0461 |

**Proposition E.4.1.** *Let* $\mathbf{a}$ *be a filter of order* $L \geq 2$. *Then, both estimators* $\hat{H}_N$ *and* $\hat{\sigma}_N$ *are strongly consistent, i.e*

$$\left( \hat{H}_N, \hat{\sigma}_N \right) \longrightarrow (H, \sigma) \quad as \quad N \longrightarrow +\infty.$$

*Moreover, we have asymptotical normality property, i.e.* $N \longrightarrow +\infty$, *for all* $H \in (0,1)$,

$$\sqrt{N} \left( \hat{H}_N - H \right) \xrightarrow{\mathcal{L}} \mathcal{N}(0, \Gamma_1(\theta, \mathbf{a}))$$

$$\frac{\sqrt{N}}{\log N} \left( \hat{\sigma}_N - \sigma \right) \xrightarrow{\mathcal{L}} \mathcal{N}(0, \Gamma_2(\theta, \mathbf{a}))$$

*where* $\Gamma_1(\theta, \mathbf{a})$ *and* $\Gamma_2(\theta, \mathbf{a})$ *symmetric definite positive matrices depending on* $\sigma$, $H$, $\alpha$ *and the filter* $\mathbf{a}$.*Brouste & Iacus (2011b)*

In the third step, utilizing the estimated values of $\hat{\sigma}$ and the $\hat{H}$ exponent, we can search for the mean reversion parameter, which is the parameter determining the rate at which the process reverts to its mean. This can be done using the following approach:

$$\hat{\alpha}_N = \left( \frac{2 \sum_{n=1}^{N} X_n^2}{\hat{\sigma}_N^2 N \Gamma \left( 2\hat{H}_N + 1 \right)} \right)^{\frac{-1}{2\hat{H}_N}} \quad .$$

Parameter estimation for the fractional Ornstein-Uhlenbeck process is a complex and intricate task. During our tests, we always had knowledge of the initial parameter set and compared the estimation results to it. By defining each parameter separately, we could observe the error step by step. It's important to note that the drift parameter estimation incorporates errors caused by previous estimations and is visible in the outcome. In practice, the fOU can be observed at discrete time points, so the selection of observation points should follow the theorem below.

**Proposition E.4.2.** *Let the observations at discrete time points* $\{t_k = kh, k = 0, 1, \ldots, n\}$. *Suppose that* $h$ *depends on* $n$ *and* $n \to \infty$, $h \to 0$ *and* $nh \to \infty$. *In addition, the following assumptions can be made on* $h$ *and* $n$ *Nualart & Nualart (2018):*

1. *When* $H \in \left(0, \frac{3}{4}\right), nh^p \to 0$ *for some* $p \in \left(1, \frac{3+2H}{1+2H} \wedge (1+2H)\right)$ *as* $n \to \infty$

2. *When* $H = \frac{3}{4}, \frac{nh^p}{\log(nh)} \to 0$ *for some* $p \in \left(1, \frac{9}{5}\right)$ *as* $n \to \infty$

3. When $H \in \left(\frac{3}{4}, 1\right), nh^p \to 0$ for some $p \in \left(1, \frac{3-H}{2-H}\right)$ as $n \to \infty$.

Table 16: Estimation of the average drift ($\alpha$) from 1000 sequences

|  |  | Sequence Length | | | | | | | |
|---|---|---|---|---|---|---|---|---|---|
| Hurst | $\alpha$ | 400 | 800 | 1600 | 3200 | 6400 | 12800 | 25600 | 51200 |
| 0.1 | 0.1 | 0.2571 | 2.2414 | 0.0000 | 0.1449 | 0.0699 | 0.2260 | 0.1740 | 0.1261 |
| 0.2 | 0.2 | 3.5961 | 1.6139 | 0.5186 | 0.4294 | 0.2692 | 0.2674 | 0.1725 | 0.1942 |
| 0.3 | 0.3 | 0.3219 | 0.1408 | 0.1984 | 0.4023 | 0.3906 | 0.4249 | 0.5208 | 0.3603 |
| 0.4 | 0.4 | 0.1260 | 0.1165 | 0.4620 | 0.6631 | 0.4218 | 0.3101 | 0.3975 | 0.4579 |
| 0.5 | 0.5 | 1.8456 | 1.1717 | 0.5133 | 0.4307 | 0.3433 | 0.4093 | 0.5899 | 0.4792 |
| 0.6 | 0.6 | 1.1175 | 0.6835 | 0.5353 | 0.5251 | 0.4033 | 0.6845 | 0.4593 | 0.5990 |
| 0.7 | 0.7 | 1.2353 | 1.7241 | 1.0004 | 0.6470 | 0.6282 | 0.7228 | 0.6592 | 0.8343 |
| 0.8 | 0.8 | 1.1798 | 1.2134 | 0.4609 | 1.1801 | 0.6363 | 0.6720 | 0.9703 | 0.7691 |
| 0.9 | 0.9 | 3.0794 | 0.4992 | 1.7640 | 1.2009 | 1.3379 | 1.8577 | 0.5694 | 0.7958 |

The generated sequences for the estimations were set up as follows: $H, \alpha \in (0.1, 0.9)$, $\sigma = 1$, where $H$ represents the Hurst exponent, $\sigma$ is the diffusion parameter, and $\alpha$ is the drift parameter. In the case of the drift and diffusion parameter, the step size was $0.1$. Additionally, we generated 1000 sequences with lengths of $400, 800, 1600, 3200, 6400, 12800, 25200, 51200$ and averaged them out. The obtained results clearly indicate that this procedure is highly sensitive to the length of the generated sequence. As the time series becomes longer, the estimation becomes more accurate. It is important to note that we set the step size $dt$ to 0.01 for each generation. The estimation results can be further improved by selecting the ideal value of $dt$ for each applied sequence length.

## F  TESTS ON SEQUENCES GENERATED BY THE YUIMA R PACKAGE

We evaluated some of our models on sequences generated by the R package YUIMA Brouste et al. (2014). In all of the cases, the evaluated models were fine-tuned for the longest sequence length in the respective tables; as such, they do not yield the best possible metrics for other sequence lengths. In every case, $10^5$ realizations were generated. In the case of the fOU and fBm, the prior distribution of the parameters was exactly as specified in Section 4. The experiments we conducted with these processes are also similar to those of Section 4. The results of these experiments can be seen in Tables 17 and 18.

Table 17: Performance metrics of fBm Hurst estimators $M_{\text{LSTM}}$ and $M_{\text{conv}}$ by sequence length on sequences generated by the YUIMA package.

|  | $MSE$ loss ($\times 10^{-3}$) | | $\hat{b}_{0.025}$ ($\times 10^{-3}$) | | $\hat{\sigma}_{0.025}$ ($\times 10^{-2}$) | |
|---|---|---|---|---|---|---|
| seq. len. | $M_{\text{LSTM}}$ | $M_{\text{conv}}$ | $M_{\text{LSTM}}$ | $M_{\text{conv}}$ | $M_{\text{LSTM}}$ | $M_{\text{conv}}$ |
| 100 | 4.78 | 5.16 | 25.7 | 19.3 | 6.12 | 6.61 |
| 200 | 2.07 | 2.19 | 12.9 | 9.73 | 4.21 | 4.41 |
| 400 | 0.947 | 1.00 | 6.10 | 4.79 | 2.92 | 3.03 |
| 800 | 0.449 | 0.470 | 3.33 | 2.47 | 2.03 | 2.10 |
| 1600 | 0.222 | 0.236 | 2.04 | 1.84 | 1.43 | 1.49 |
| 3200 | 0.111 | 0.121 | 1.46 | 1.88 | 1.011 | 1.06 |
| 6400 | 0.0565 | 0.0668 | 1.35 | 2.68 | 0.715 | 0.748 |
| 12800 | 0.0300 | 0.0412 | 1.40 | 3.07 | 0.511 | 0.535 |

Table 18: Performance metrics of fOU Hurst estimator $M_{\text{LSTM}}$ by sequence length on sequences generated by the YUIMA package.

| seq. len. | MSE ($\times 10^{-3}$) | $\hat{b}_{0.025}$ ($\times 10^{-3}$) | $\hat{\sigma}_{0.025}$ ($\times 10^{-2}$) |
|---|---|---|---|
| 100 | 8.19 | 16.0 | 8.37 |
| 200 | 2.85 | 7.23 | 5.08 |
| 400 | 1.22 | 4.13 | 3.35 |
| 800 | 0.577 | 2.79 | 2.32 |
| 1600 | 0.293 | 2.19 | 1.65 |
| 3200 | 0.151 | 1.94 | 1.17 |
| 6400 | 0.0815 | 2.25 | 0.839 |

