# OpenReview forum: "Parameter Estimation of Long Memory Stochastic Processes with Deep Neural Networks"
_ICLR.cc/2024/Conference — Submitted to ICLR 2024_

### Official Review · Reviewer_H3eJ · 2023-10-30

**Soundness:** 3 good
**Presentation:** 1 poor
**Contribution:** 3 good
**Rating:** 5
**Confidence:** 3

**Summary:**

The paper studies the problem of estimating parameters of long stochastic processes with long-range dependencies. The proposed method generates high-quality synthetic training data to train neural networks that are able to capture the long-range dependencies in the data. The paper experimentally demonstrates the benefits of their method compared to a set of baselines.

**Strengths:**

The paper studies a very relevant problem of learning long-range dependencies and estimating involved parameters. The key novelty I see in this work is the relation of the dependency in the data and the estimation of parameters to capture it.

**Weaknesses:**

- The paper misses significant breakthroughs in the domain of learning long-term dependencies using neural networks. In particular, [1,2,3,4] pushed the boundary on the sequence length of data that can be learned by an ML model. It is important to see how the proposed methods work compared to and in conjunction with these approaches.
- The writing of the paper can be improved. In particular, it is unclear what is new and what is just standard training methods. I think the paper would be much stronger if the contributions and relations to existing methods were stated more clearly.

**References**.
[1] Gu et al. 2021, Efficiently Modeling Long Sequences with Structured State Spaces.
[2] Rusch et al. 2021, Unicornn: A recurrent model for learning very long time dependencies.
[3] Morrill et al. 2021, Neural Rough Differential Equations for Long Time Series
[4] Rusch et al. 2022, Long expressive memory for sequence modeling.

**Questions:**

Can't the stochastic process parameters in the paper much better be characterized using the log-signature? (e.g. see Morrill et al. 2021)

---

> ### Author Response · Authors · 2023-11-23
>
> Thank you very much for the review. We would like to make the following observations.
>
> > The paper misses significant breakthroughs in the domain of learning long-term dependencies using neural networks. In particular, [1,2,3,4] pushed the boundary on the sequence length of data that can be learned by an ML model. It is important to see how the proposed methods work compared to and in conjunction with these approaches.
>
> In our opinion, the problem of parameter estimation is not the same as the (general) task of modeling long sequences. While in the latter case, the proposed articles are clearly relevant, for the problem studied in our article, neural networks processing the raw sequences can produce similar or better results. Nevertheless, in the introduction the paper Kidger et al. (2019) is also cited. In practice, the usable sequence length ranges from a few hundred to a few thousand, e.g. due to the frequently occurring time dependence of the parameters.
>
> > The writing of the paper can be improved. In particular, it is unclear what is new and what is just standard training methods.
>
> The new insight provided by the article is the realization that sequence processing neural architectures trained on a sufficient amount of data, can in itself replace the role of even sophisticated statistical descriptors in determining the memory parameters characterizing processes, somewhat similarly to the role of convolutional networks in image processing. This is formulated in the Introduction and in full concreteness in the Conclusion. The main part of the measurements is to argue on the power of these latent calculations leading to the parameters.
>
> > Can't the stochastic process parameters in the paper much better be characterized using the log-signature? (e.g. see Morrill et al. 2021)
>
> The advantage of signature-based approaches comes in the case of multi-dimensional series. In our case, the input data is one-dimensional, and the relations of the stochastic series are considered to be independent of each other.
> Thus, the use of the signature will not give you more than the use of the spectrum domain, while it has a fairly significant calculation requirement. On the one hand, we carried out specific measurements according to Kidger et al. (2019), and we also took the raw Fourier spectrogram, which was then processed with MLP or LSTM. These measurements also show that it is not possible to achieve a better result than the sequence processing neural networks working on raw sequences.

---

### Official Review · Reviewer_Rt5S · 2023-11-01

**Soundness:** 2 fair
**Presentation:** 2 fair
**Contribution:** 2 fair
**Rating:** 3
**Confidence:** 3

**Summary:**

The authors propose a method for fitting fractional Brownian motion using neural network sequence models. They test the method by creating many simulated datasets of varying lengths and assessing the each model's quality of fit with MSE on the observation, analysis of bias and deviation, recovery of the Hurst parameter.

**Strengths:**

The results seem to suggest that this method does in fact work better than traditional alternatives to model fitting on fBms and by a significant margin.

**Weaknesses:**

I'm not an expert in this domain, but it was very challenging to tell what was actually being proposed in this paper.

My first interpretation was that the parameters of the fBm were parameterized and then a differentiable integration method was applied to the fBM to get sample paths which were compared with ground truth paths using MSE. Then the parameters could be found by backpropagating through the integration to the parameters. But naively this would not require a sequence model, which is describe to take in a sequence and output a single scalar through mean aggregation. It's not obvious to me what the neural network is actually being used for. Is it also being used to integrate the process in some way? The presentation could be improved significantly with appropriate explanatory figures or at least a description of the actual training loss and simulation procedure.

**Questions:**

Is it possible the baselines are too weak? They seem to be *significantly* worse than a straightforward application of sequence models. It's a little hard to believe there aren't other deep learning methods that could be used as baselines here. There is has been substantial work on learning SDEs (e.g. Patrick Kidger's work) and other stochastic processes with neural networks (e.g. neural diffusion processes) and those methods might be applicable here.

What is the intended use case for this model? In the paper it states that the setup implies there is infinite data. I see how this is true when fitting models to synthetically generated data, but it's almost certainly not true when attempting to fit real world data. If this is meant to be used in the types of financial or scientific applications described in the introduction to the paper, why not apply it directly to those applications and evaluate how fell it performs in terms of MSE?

---

> ### Author Response · Authors · 2023-11-23
>
> Thank you very much for the review. We add to the summary that a large part of the paper reflects on fBm, however, two other important processes, ARFIMA and fOU, are also in focus in the article. We are happy to read the review’s mentioning that the selected benchmarks were significantly exceeded by our approach.
>
> > My first interpretation was that the parameters of the fBm were parameterized and then a differentiable integration method was applied to the fBM to get sample paths which were compared with ground truth paths using MSE. Then the parameters could be found by backpropagating through the integration to the parameters. But naively this would not require a sequence model, which is describe to take in a sequence and output a single scalar through mean aggregation. It's not obvious to me what the neural network is actually being used for. Is it also being used to integrate the process in some way? The presentation could be improved significantly with appropriate explanatory figures or at least a description of the actual training loss and simulation procedure.
>
> Regarding the content of the article, it is advisable to differentiate between
> (1) modeling long sequences, (2) modeling and forecasting processes with a long memory, (3) measuring how much memory a short or long sequence has, also covering the case when the sequence is stochastic.
> In the paper, we dealt with the third case, where we chose three stochastic processes used in practice and their equidistant realizations as models. The parameters characterizing the memory (or fractal dimension) are estimated directly from the raw sequence data, using an LSTM architecture. We feel that the self-standing use of the neural network for this task is novel, especially in light of the achieved achievements. It is also new to cover the scaling problem of the processes.
>
> > Is it possible the baselines are too weak? They seem to be significantly worse than a straightforward application of sequence models. It's a little hard to believe there aren't other deep learning methods that could be used as baselines here. There is has been substantial work on learning SDEs (e.g. Patrick Kidger's work) and other stochastic processes with neural networks (e.g. neural diffusion processes) and those methods might be applicable here.
>
> In our opinion, the baselines have been established carefully, based on the literature and very broad practice. As mentioned in the introduction, we looked at several (more or less well-described and supported) methods combining neural network methods with statistical descriptors. Among them was Kidger et al. (2019) signatures method. We found that the use of predefined statistical descriptors in the parameter estimation task could be offset by teaching a large amount of data.
>
> > What is the intended use case for this model? In the paper it states that the setup implies there is infinite data. I see how this is true when fitting models to synthetically generated data, but it's almost certainly not true when attempting to fit real world data. If this is meant to be used in the types of financial or scientific applications described in the introduction to the paper, why not apply it directly to those applications and evaluate how fell it performs in terms of MSE?
>
> The reviewer's reasonable question is why it is sufficient to create a model for the class of stochastic processes often used in modeling real processes. When real sequential data is inferred by these models, the data is fit to the closest element of the chosen stochastic class measured in MLE. If we want to produce a model with another stochastic class, we train a neural network for the parameters of this next class. This is what happens in the case of the ARFIMA and fOU processes presented in the article. In this way, we avoid teaching on real data limited by the smaller volume of data and such phenomena as e.g. the Hurst parameter changes (in time) through a real data series. This also makes the evaluation on real series relative. Nevertheless, we have some measurements when we infer with our Hurst estimation model on real data series, and the obtained outputs are consistent with other results, e.g. with baseline methods.
>
> Teaching on independent records of large volume is an important factor. It requires a sufficiently efficient process generator, which produces an unlimited amount of usable teaching data, which we call infinite, with some exaggeration.

---

### Official Review · Reviewer_pWyU · 2023-11-01

**Soundness:** 3 good
**Presentation:** 2 fair
**Contribution:** 1 poor
**Rating:** 3
**Confidence:** 3

**Summary:**

Fractional Brownian motion, Autoregressive Fractionally Integrated Moving Average and the fractional Ornstein-Uhlenbeck process are often used in real world. They are governed by the Hurst or differencing parameter which this paper estimates using neural networks. Having generated many different samples the network learns to output the true underlying parameter.

**Strengths:**

The paper is easy to follow. The baselines established in 2.4 are reasonable. The approach is simple but it is well motivated.

**Weaknesses:**

The scope of the problem is very limited. Perhaps showing that this approach scales to different equations at once, or having a *foundation* model for symbolic regression.

The results in Figure 1 (right) are not that impressive. An interesting contribution would be to have a model that scales from short sequences to large ones.

I believe that the novelty, significance and the results are not enough for this conference.

**Questions:**

.

---

> ### Author Response · Authors · 2023-11-23
>
> Thank you very much for the review. We appreciate the reviewer’s feedback on the correctness of the methodology and baseline selection.
>
> > The scope of the problem is very limited. Perhaps showing that this approach scales to different equations at once, or having a foundation model for symbolic regression.
>
> We agree that many interesting questions beyond the results of this paper can be studied from the topic of parameter estimation of fractional processes with neural networks. We provide several examples of that in the Appendix of the article. Nevertheless, we believe that it was important to demonstrate the pure neural network parameter estimation approach in such a way that it can be used in practice, is easy to overview, and thus appears as a method accessible to many.
>
> Thank you for the idea of a neural estimation method that works accurately for multiple processes at the same time, which is an exciting suggestion. In many respects, the stress tests described in Appendix D can be considered steps in this direction. The connection to symbolic regression is indeed natural, but limiting the computing power of neural networks for the sake of transparent calculations can lead to a significant loss of accuracy. In this article, we clearly wanted to focus on the performance of the methods.
>
> > The results in Figure 1 (right) are not that impressive. An interesting contribution would be to have a model that scales from short sequences to large ones.
>
> As for Figure 1 (right), we cannot use short series as input for the parameter estimation beyond a certain level of accuracy (see e.g. Weron 2002, Physica A 312). On the other hand, incremental learning with increasingly long sequences preserves the ability to estimate with shorter sequences.

---

### Official Review · Reviewer_4CiJ · 2023-11-10

**Soundness:** 2 fair
**Presentation:** 2 fair
**Contribution:** 1 poor
**Rating:** 3
**Confidence:** 3

**Summary:**

This paper introduces the use of efficient process generators to estimate the long range parameters of stochastic process models via a purely deep neural network approach that does not use conventional statistical methods. Background information on these time series that exhibit long range dependence is provided and some experimental results are supplied to validate the approach.

**Strengths:**

The main strengths of this paper are quite limited and therefore, I have spent more time in highlighting the weaknesses and shortcomings of this paper.

**Weaknesses:**

The major novel contribution of this work is not clearly specified at all. The paper has the following weaknesses:

1) This work seems to be an application of available process generators to generate high quality synthetic data for the fBm and other long range stochastic processes to train standard neural network models and evaluate their performance. This contribution is not sufficient for an ICLR paper.

2) The paper reads like a background on stochastic processes exhibiting long range dependence and a small section devoted to the actual tasks implemented by the authors. It does not make for good reading.

**Questions:**

The line of work is interesting, and I urge the authors to undertake more detailed theoretical analysis and possibly even some guarantees on estimating these long range memory parameters using recurrent neural networks and other architectures.

---

> ### Author Response · Authors · 2023-11-23
>
> Thank you very much for the review. We would like to make the following comments.
>
> > The major novel contribution of this work is not clearly specified at all.
>
> The new insight provided by the article is the realization that sequence processing neural architectures, trained on a sufficient amount of data, can in itself replace even sophisticated statistical descriptors in determining the memory parameters characterizing processes, somewhat similarly to the role of convolutional networks in image processing. This is formulated in the Introduction and, in full concreteness, in the Conclusion. The main part of the measurements is to argue on the power of these latent calculations leading to the parameters.
>
> > This work seems to be an application of available process generators to generate high quality synthetic data for the fBm and other long range stochastic processes to train standard neural network models and evaluate their performance.
>
> > The paper reads like a background on stochastic processes exhibiting long range dependence and a small section devoted to the actual tasks implemented by the authors.
>
> Our result does indeed operate with existing tools, but it shows many previously unknown or unachieved insights. Obviously, the most important thing is the accuracy and speed of parameter extraction; the superior nature of our method is discussed in detail in Section 4.
> The question also arises whether it is possible to show better performance if we do not work directly from the raw data of realization but rather extract some complex features, e.g. statistical descriptors, and use them as input for a neural network. In this regard, we have carried out several measurements, and neither the use of spectrum domain nor the signatures give better performance while they require significantly more computing resources.
> We note that the codes used for measurements will be available, which also indicates that the paper is more than a compilation of off-the-shelf methods.
>
> > The line of work is interesting, and I urge the authors to undertake more detailed theoretical analysis and possibly even some guarantees on estimating these long range memory parameters using recurrent neural networks and other architectures.
>
> In response to the formulated question, we are happy to have received confirmation regarding the interest in the topic. Motivated by the suggestion, we are thinking about which mathematical-statistical methods could be used to achieve analytical results, e.g. lower estimate of achievable accuracy.
>
> Finally, we would like to remark that the title of the article draws attention to long-memory processes, however, the article is also about short-memory or rough processes (e.g. fBm with Hurst near 0).

---

### Meta-Review · Area_Chair_8cKK · 2023-12-08

**Metareview:**

The paper presents efficient process generators, employing a deep neural network approach to estimate long-range parameters in stochastic process models without traditional statistical methods. It focuses on key processes, estimating their governing parameters through neural networks and testing the proposed method on simulated datasets.

The weaknesses identified revolve around the lack of clarity in the paper's novel contribution, limited problem scope, unimpressive results, unclear presentation, and a failure to acknowledge and compare with significant breakthroughs in the field. Addressing these concerns is crucial for enhancing the paper's overall quality and acceptance in the conference.

**Justification For Why Not Higher Score:**

The weaknesses identified revolve around the lack of clarity in the paper's novel contribution, limited problem scope, unimpressive results, unclear presentation, and a failure to acknowledge and compare with significant breakthroughs in the field. Addressing these concerns is crucial for enhancing the paper's overall quality and acceptance in the conference.

**Justification For Why Not Lower Score:**

N/A

---

### Decision · Program_Chairs · 2024-01-16

Reject